# Kernel Banzhaf: A Fast and Robust Estimator for Banzhaf Values

## Abstract

Banzhaf values offer a simple and interpretable alternative to the widely-used Shapley values. We introduce Kernel Banzhaf, a novel algorithm inspired by KernelSHAP, that leverages an elegant connection between Banzhaf values and linear regression. Through extensive experiments on feature attribution tasks, we demonstrate that Kernel Banzhaf substantially outperforms other algorithms for estimating Banzhaf values in both sample efficiency and robustness to noise. Furthermore, we prove theoretical guarantees on the algorithm's performance, establishing Kernel Banzhaf as a valuable tool for interpretable machine learning.

## 1 Introduction

The increasing complexity of AI models has intensified the challenges associated with model interpretability. Modern machine learning models, such as deep neural networks and complex ensemble methods, often operate as "opaque boxes." This opacity makes it difficult for users to understand and trust model predictions, especially in decision-making scenarios like healthcare, finance, and legal applications, which require rigorous justifications. Thus, there is a pressing need for reliable explainability tools to bridge the gap between complex model behaviors and human understanding.

Among the various methods employed within explainable AI, game-theoretic approaches have gained prominence for quantifying the contribution of features in predictive modeling and enhancing model interpretability. While primarily associated with feature attribution (Lundberg & Lee, 2017; Karczmarz et al., 2022), these methods also contribute to broader machine learning tasks such as feature selection (Covert et al., 2020) and data valuation (Ghorbani & Zou, 2019; Wang & Jia, 2023). Such applications extend the utility of explainable AI, fostering greater trust in AI systems by providing insights beyond traditional explanations.

Shapley values, rooted in cooperative game theory, provide a principled way to attribute the contribution of $n$ individual players to the overall outcome of a game (Shapley, 1953). In the context of feature attribution, each "player" is a feature and the "game" is defined by a *set function* that maps a subset of features to the prediction score of an AI model on that subset. For a given feature, the Shapley value quantifies the average marginal contribution of the feature on the model's prediction, computed as the weighted average over all possible combinations of features included in the model (see e.g., Equation 1) (Lundberg & Lee, 2017).

An alternative to Shapley values are Banzhaf values, which also compute each individual's contribution to the system's overall outcome (Banzhaf, 1965). While Shapley values are more widely used, Banzhaf values are often considered more intuitive for AI applications since they treat each subset of players as equally important, directly measuring the impact of each player across all possible combinations (see e.g., Equation 2). Additionally, Banzhaf values can be computed more efficiently and tend to be more numerically robust (Karczmarz et al., 2022; Wang & Jia, 2023).

For general set functions, the exact computation of Shapley and Banzhaf values is an NP-hard problem (Deng & Papadimitriou, 1994), so they are often approximated in practice. The problem of estimating Shapley values, especially for model explanation, has been well-studied. A leading method is KernelSHAP (Lundberg & Lee, 2017), a model-agnostic technique that leverages a connection to linear regression, approximating Shapley values by solving a subsampled weighted least squares problem (Charnes et al., 1988; Lundberg & Lee, 2017). KernelSHAP has been further improved with paired sampling (Covert & Lee, 2020) and leverage score sampling Musco & Witter (2024).

In contrast to Shapley values, only a few algorithms have been proposed to compute Banzhaf values for arbitrary set functions. For tree-structured set functions, such as decision tree based models,

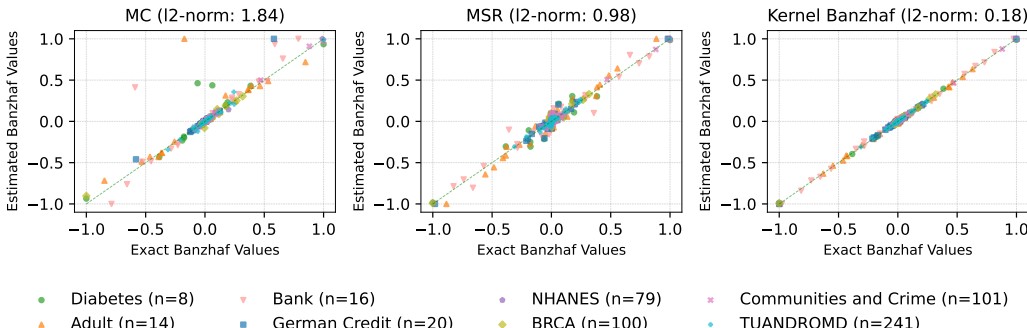

Figure 1: Comparison of exact and estimated Banzhaf values across datasets and estimators using $20n$ samples. Each subplot, labeled with its estimator, shows the normalized estimated versus exact Banzhaf values across all features for a randomly selected explicand from each dataset. Points closer to the diagonal line indicate more accurate estimates; the qualitative evaluation and the $\ell_2$-norm error indicate that Kernel Banzhaf is more accurate than the MC and MSR estimators.

exact Banzhaf values can be efficiently computed (Karczmarz et al., 2022). For general set functions (e.g., opaque neural networks), Monte Carlo sampling can be used to estimate each Banzhaf value separately (Bachrach et al., 2010). The Maximum Sample Reuse (MSR) algorithm reuses samples for the estimates of different Banzhaf values (Wang & Jia, 2023). However, MSR estimates are correlated and have larger magnitude, and hence high variance as discussed in Section 4.1.

In this work, we apply ideas that have been proven effective for estimating Shapley values to Banzhaf values (Lundberg & Lee, 2017; Covert & Lee, 2020; Musco & Witter, 2024). Our starting point is a formulation of Banzhaf values in terms of a specific linear regression problem (Section 3.1). While a connection to linear regression was known for 'simple' set functions (i.e., set functions with binary outputs that satisfy a monotonicity property) in Hammer & Holzman (1992), to the best of our knowledge, we are the first to establish the connection for general set functions. We leverage this connection to design a new algorithm for approximating Banzhaf values: we propose Kernel Banzhaf, inspired by KernelSHAP, to solve a subsampled instance of the Banzhaf linear regression problem (Section 3.2). Similar to Leverage SHAP (Musco & Witter, 2024), we exploit leverage score sampling to give approximation guarantees on the $\ell_2$-norm error.

We show that with $O(n \log \frac{n}{\delta} + \frac{n}{\delta \epsilon})$ samples, Kernel Banzhaf produces estimated Banzhaf values $\hat{\phi}$ such that $\|\hat{\phi} - \phi\|_2^2 \leq \epsilon \gamma \|\phi\|_2^2$ with probability $1 - \delta$, where $\phi$ are the exact Banzhaf values.[1] In contrast, Wang & Jia (2023) show that with $O(\frac{n}{\epsilon} \log \frac{n}{\delta})$ samples, MSR produces estimates such that $\|\hat{\phi}^{\mathrm{MSR}} - \phi\|_2^2 \leq \epsilon$. However, their result assumes that the set function output is restricted to $[0, 1]$, excluding regression and generative AI tasks.[2]

Beyond theoretical guarantees, we extensively evaluate the empirical performance of Kernel Banzhaf. Our experimental results show that Kernel Banzhaf produces estimates which are closer to the true Banzhaf values than MC and MSR (Figure 1). In prior work, estimators for Banzhaf values on large datasets were compared in terms of their convergence (Wang & Jia, 2023). Although convergence is an important property, it does not guarantee accuracy, particularly if the estimators have not been proven to be unbiased. In our work, we evaluate the accuracy of Banzhaf estimators for large datasets by using the algorithms of Karczmarz et al. (2022) to compare against exact values. Our findings, across eight datasets, suggest that Kernel Banzhaf is more robust and accurate than prior estimators. Additionally, we compare Kernel Banzhaf with state-of-the-art KernelSHAP algorithms and empirically demonstrate that in semivalue estimation, Kernel Banzhaf provides robustness across all scales and greater efficiency particularly when dealing with a large number of features. We further substantiate this finding by examining the condition number of these regression-based methods.

**Contributions** Our main contributions can be summarized as follows:

---

[1] We believe that the $\gamma$ parameter is necessary for linear regression-based algorithms (see Section 3.3).

[2] Because the upper bound of their guarantee has no dependence on $\|\phi\|_2^2$, their assumption on the range of the set function effectively normalizes the set function to satisfy the guarantee.

1. We propose Kernel Banzhaf, a regression-based approximation algorithm for estimating Banzhaf values for general set functions.

2. We show that Kernel Banzhaf provably returns accurate solutions. Our analysis requires non-trivial modifications to the standard leverage score sampling analysis. We argue that, up to log factors and the dependence on $\epsilon$, our analysis is the best possible.

3. We run a detailed experimental evaluation using eight popular datasets to evaluate the performance of algorithms for approximating Banzhaf values and Shapley values in feature attribution tasks, varying the number of samples and the reliability of the set function to simulate practical conditions. Unlike prior work, our experiments evaluate estimators relative to the true Banzhaf values, rather than relying just on convergence metrics. Our findings demonstrate that Kernel Banzhaf significantly outperforms existing methods in efficiency and robustness.

## 2 BACKGROUND

Let $n$ be the number of players and $v : 2^{[n]} \to \mathbb{R}$ be a set function. The Shapley value of player $i$ for $i \in [n] := \{1, \ldots, n\}$ is

$$\phi_i^{\text{Shapley}} = \frac{1}{n} \sum_{S \subseteq [n] \setminus \{i\}} \binom{n-1}{|S|}^{-1} [v(S \cup \{i\}) - v(S)] \tag{1}$$

while the Banzhaf value is

$$\phi_i^{\text{Banzhaf}} = \frac{1}{2^{n-1}} \sum_{S \subseteq [n] \setminus \{i\}} [v(S \cup \{i\}) - v(S)] \tag{2}$$

Shapley values and Banzhaf values each uniquely satisfy four different properties. We defer the description of these properties to Appendix H. By default, we shall use $\phi$ to denote $\phi^{\text{Banzhaf}}$ unless explicitly stated otherwise and use $\phi \in \mathbb{R}^n$ to denote the vector $[\phi_1, \ldots, \phi_n]$.

We can recover various tasks in machine learning by choosing different ways of defining the set function. In this work, we will consider one of the most popular tasks in explainable AI: feature attribution (also known as feature importance and feature influence). Let $M$ be some model which takes in input $\mathbf{x} \in \mathbb{R}^n$. In the version we consider, the feature attribution seeks to explain the predictions of the model on an *explicand* observation $\mathbf{x}^e$, which is the target data point for which we want to understand the contributions of individual features. Given a subset of features $S$, define $\mathbf{x}^S$ as the observation where $\mathbf{x}_i^S = \mathbf{x}_i^e$ if feature $i \in S$ and, otherwise, $\mathbf{x}^{\bar{S}}$ is sampled from a distribution, as discussed in Appendix B. $v(S) = \mathbb{E}[M(\mathbf{x}^S)]$, where the expectation is over the sampling of the features $i \notin S$. We can adapt the algorithm to other machine learning tasks such as feature (or data) selection by redefining $v(S)$ as the loss of a model trained using only the features (or observations) within subset $S$.

## 3 KERNEL BANZHAF

The starting point of our work is a formulation of Banzhaf values in terms of a specially structured linear regression problem, and in Theorem 3.2, we show that Banzhaf values are the exact solution to this linear regression problem. Then we propose the *Kernel Banzhaf* algorithm for estimating Banzhaf values.

### 3.1 LINEAR REGRESSION FORMULATION

Consider binary vectors $\mathbf{z} \in \{-1, 1\}^n$. In a slight abuse of notation, we will use $\mathbf{z}$ as input to the set function where the corresponding set is induced by $\mathbf{z}$. We will also use $\mathbf{z}$ as an index for the following matrices.

- Let $\mathbf{A} \in \mathbb{R}^{2^n \times n}$ where the row corresponding to $\mathbf{z}$ is given by $[\mathbf{A}]_{\mathbf{z}} = \frac{1}{2}\mathbf{z}^\top$.

- Let $\mathbf{b} \in \mathbb{R}^{2^n}$ where the entry corresponding to $\mathbf{z}$ is given by $\mathbf{b}_{\mathbf{z}} = v(\mathbf{z})$.

The following observation about $\mathbf{A}$ will be helpful in our analysis.

**Observation 3.1.** $\mathbf{A}^\top \mathbf{A} = 2^{n-2}\mathbf{I}$

*Proof.* The $(i, j)$ entry in $\mathbf{A}^\top \mathbf{A}$ is given by

$$[\mathbf{A}^\top \mathbf{A}]_{i,j} = \sum_{\mathbf{z}} \frac{1}{4} z_i z_j \tag{3}$$

If $i \neq j$, there are $2^{n-1}$ terms of $-\frac{1}{4}$ when $z_i \neq z_j$ and $2^{n-1}$ terms of $+\frac{1}{4}$ when $z_i = z_j$ hence Equation 3 is 0. If $i = j$, we have $2^n$ terms of $+\frac{1}{4}$ hence Equation 3 is $2^{n-2}$. Together, this gives that $\mathbf{A}^\top \mathbf{A} = 2^{n-2}\mathbf{I}$. $\qquad\square$

With this observation, we will establish that Banzhaf values are the solution to the linear regression problem defined on $\mathbf{A}$ and $\mathbf{b}$. A similar result was known but only for the case where $v$ is a simple set function (the output is binary $\{0, 1\}$ and the function satisfies a monotonicity property) (Hammer & Holzman, 1992).

**Theorem 3.2** (Linear Regression Equivalence). *The Banzhaf values are the solution to the linear regression problem induced by $\mathbf{A}$ and $\mathbf{b}$, i.e.,*

$$\phi = \arg\min_{\mathbf{x}} \|\mathbf{A}\mathbf{x} - \mathbf{b}\|_2. \tag{4}$$

The proof shows that the optimal linear regression solution is equal to the Banzhaf values.

*Proof.* By setting the gradient of the objective function to $\mathbf{0}$, we have that

$$\arg\min_{\mathbf{x}} \|\mathbf{A}\mathbf{x} - \mathbf{b}\|_2 = (\mathbf{A}^\top \mathbf{A})^{-1} \mathbf{A}^\top \mathbf{b}. \tag{5}$$

We will analyze the right hand side. By Observation 3.1, we have $\mathbf{A}^\top \mathbf{A} = 2^{n-2}\mathbf{I}$. Then $(\mathbf{A}^\top \mathbf{A})^{-1} = \frac{1}{2^{n-2}}\mathbf{I}$. Continuing, we have

$$(\mathbf{A}^\top \mathbf{A})^{-1} \mathbf{A}^\top \mathbf{b} = \frac{1}{2^{n-2}} \mathbf{A}^\top \mathbf{b} = \frac{1}{2^{n-2}} \sum_{\mathbf{z}} \frac{1}{2} \mathbf{z} v(\mathbf{z}) = \frac{1}{2^{n-1}} \sum_{\mathbf{z}} \mathbf{z} v(\mathbf{z}). \tag{6}$$

Now consider entry $i$ given by

$$[(\mathbf{A}^\top \mathbf{A})^{-1} \mathbf{A}^\top \mathbf{b}]_i = \frac{1}{2^{n-1}} \sum_{\mathbf{z}} z_i v(\mathbf{z}) = \frac{1}{2^{n-1}} \sum_{S \subseteq [n] \setminus \{i\}} v(S \cup \{i\}) - v(S) \tag{7}$$

which is exactly Equation 2. The statement follows. $\qquad\square$

## 3.2 THE KERNEL BANZHAF ALGORITHM

Since $\mathbf{A}$ and $\mathbf{b}$ are exponentially large, constructing the linear regression problem to calculate the Banzhaf values is computationally prohibitive. Instead, we construct a smaller linear regression problem with $m$ samples. Let the subsampled matrix be $\tilde{\mathbf{A}} \in \mathbb{R}^{m \times n}$ and the target vector be $\tilde{\mathbf{b}} \in \mathbb{R}^m$. The estimate we produce is

$$\hat{\phi} = \arg\min_{\mathbf{x}} \|\tilde{\mathbf{A}}\mathbf{x} - \tilde{\mathbf{b}}\|_2. \tag{8}$$

Building on this framework, we introduce the *Kernel Banzhaf* algorithm, which utilizes paired sampling and leverage score sampling to construct the small regression problem (Woodruff et al., 2014; Drineas & Mahoney, 2018). The leverage scores of $\mathbf{A}$ are given by

$$\ell_{\mathbf{z}} = [\mathbf{A}]_{\mathbf{z}} (\mathbf{A}^\top \mathbf{A})^{-1} [\mathbf{A}]_{\mathbf{z}}^\top = \frac{1}{2^{n-2}} \left(\frac{1}{2}\mathbf{z}^\top\right) \left(\frac{1}{2}\mathbf{z}\right) = \frac{n}{2^n} \tag{9}$$

where the third equality follows by Observation 3.1. Since the leverage scores are the same for all rows, uniform sampling is equivalent to sampling by leverage scores for $\mathbf{A}$. As such, Kernel Banzhaf selects $\tilde{\mathbf{A}}$ and $\tilde{\mathbf{b}}$ by sampling each row uniformly at random with replacement.

---

**Algorithm 1** Kernel Banzhaf

---

**Input:** Set function $v : \{0, 1\}^n \to \mathbb{R}$, $n$ players, $m$ samples s.t. $n < m \leq 2^n$.
**Output:** Approximate Banzhaf values $\hat{\phi} \in \mathbb{R}^n$
 1: $\mathbf{Z} \leftarrow \mathbf{0}_{m \times n}$
 2: **for** $i = 0, \ldots, \lfloor m/2 \rfloor - 1$ **do**
 3:    $\mathbf{z} \leftarrow$ random binary vector sampled uniformly
 4:    $\bar{\mathbf{z}} \leftarrow$ complement of $\mathbf{z}$              $\triangleright$ I.e., $\bar{z}_j = 1 - z_j$ for $0 \in \{0, \ldots, n - 1\}$
 5:    $\mathbf{Z}_{2i} \leftarrow \mathbf{z}^\top$, $\mathbf{Z}_{2i+1} \leftarrow \bar{\mathbf{z}}^\top$
 6: **end for**
 7: $\tilde{\mathbf{b}} \leftarrow [v(\mathbf{Z}_1), \ldots, v(\mathbf{Z}_m)]$
 8: $\tilde{\mathbf{A}} \leftarrow \mathbf{Z} - \frac{1}{2}\mathbf{1}_{m \times n}$                   $\triangleright$ Convert from $\{0, 1\}$ to $\{-\frac{1}{2}, \frac{1}{2}\}$
 9: $\hat{\phi} \leftarrow \arg\min_{\mathbf{x}} \|\tilde{\mathbf{A}}\mathbf{x} - \tilde{\mathbf{b}}\|_2$              $\triangleright$ Standard least squares
10: **return** $\hat{\phi}$

---

The pseudocode for Kernel Banzhaf appears in Algorithm 1.

We now consider the time complexity of Algorithm 1. The time to randomly sample with replacement is $O(n)$ per sample, along with the time to compute the complement and store both vectors. Let $T_m$ be the time complexity to evaluate the set function $m$ times, possibly in parallel. This cost is unavoidable for any algorithm that evaluates the set function $m$ times. Finally, the least squares regression problem requires $O(mn^2)$ time because the matrix $\tilde{\mathbf{A}}$ has $m$ rows and $n$ columns and the vector $\tilde{\mathbf{b}}$ has $m$ entries. In total, the time complexity of Algorithm 1 is $O(T_m + mn^2)$. For most set functions, we expect the $O(T_m)$ term to dominate. For example, even a forward pass on a shallow fully connected neural network with dimension at most $n$ will take $O(n^2)$ time per sample.

The state-of-the-art methods for Shapley estimation, KernelSHAP (Lundberg & Lee, 2017; Covert & Lee, 2020) and Leverage SHAP (Musco & Witter, 2024), sample without replacement to enhance performance when $m \approx n$. Our empirical results in Figure 10 show no improvements with this approach (until $m = 2^n$, in which case we can exactly compute Banzhaf values). Moreover, sampling without replacement exhibits lower efficiency and scalability, and it does not ensure the use of exactly $m$ samples. Further details are provided in Appendix G.

### 3.3 Approximation Guarantees

**Theorem 3.3.** *If $m = O(n \log \frac{n}{\delta} + \frac{n}{\delta\epsilon})$, Algorithm 1 produces an estimate $\hat{\phi}$ that satisfies*

$$\|\mathbf{A}\hat{\phi} - \mathbf{A}\phi\|_2 \leq (1 + \epsilon)\|\mathbf{A}\phi - \mathbf{b}\|_2 \tag{10}$$

*with probability $1 - \delta$.*

Theorem 3.3 is a standard guarantee for leverage score sampling. However, establishing the theorem for Kernel Banzhaf requires completely reproving it from scratch because of the incorporation of *paired sampling* in Kernel Banzhaf. For the analysis of Leverage SHAP, Musco & Witter (2024) prove a similar theorem with paired leverage score sampling but using different techniques, since the samples in their algorithm are taken without replacement.

The theorem can only be improved in the logarithmic factor and dependence on $\delta$ and $\epsilon$. To see why, consider a linear set function where $\mathbf{b}$ is in the span of $\mathbf{A}$. Then the right-hand side is 0 and we must recover $\phi$ exactly. To do this, we need to observe at least $n$ linearly independent rows hence we need at least $\Omega(n)$ samples.

While it gives a strong guarantee, the term in Theorem 3.3 is less interpretable. Fortunately, we can use the special properties of $\mathbf{A}$ and $\mathbf{b}$ to prove the following corollary in terms of the $\ell_2$-norm error.

**Corollary 3.4.** *Define $\gamma = \|\mathbf{A}\phi - \mathbf{b}\|_2^2 / \|\mathbf{A}\phi\|_2^2$. Any $\hat{\phi}$ that satisfies Equation 10 also satisfies*

$$\|\hat{\phi} - \phi\|_2^2 \leq \epsilon\gamma\|\phi\|_2^2. \tag{11}$$

The proof is similar to guarantees for general linear regression problems and for the Leverage SHAP setting (see e.g., Lemma 68 in Drineas & Mahoney (2018) and Corollary 4.1 in Musco & Witter

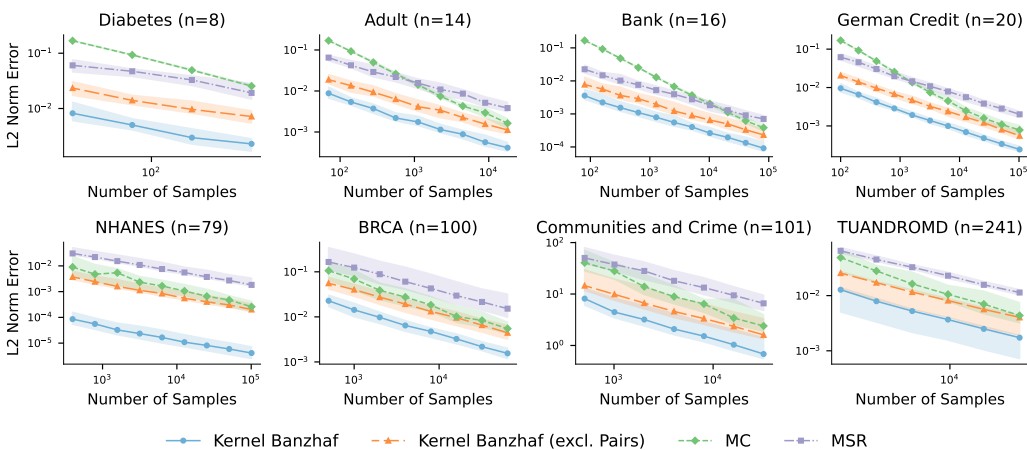

Figure 2: Plots comparing the $\ell_2$-norm errors (i.e. $\|\hat{\phi} - \phi\|_2^2$) for Kernel Banzhaf (and its ablated version without paired sampling), MC, and MSR across increasing sample sizes in eight datasets. Each point represents the median of 50 runs, with shaded areas indicating the 25th to 75th percentiles. The plots showcase the effectiveness of Kernel Banzhaf in various real-world datasets.

(2024)). However, because of the special structure of our problem, our result differs in that there is no relaxation in the guarantee from Theorem 3.3 to Corollary 3.4. Hence the (near-)optimality of Theorem 3.3 implies the (near-)optimality of Corollary 3.4.

Notice that Theorem 3.3 has no dependence on $\gamma$ while Corollary 3.4 does. This suggests that, when $\mathbf{b}$ is outside the span of $\mathbf{A}$, Kernel Banzhaf can recover a solution $\hat{\phi}$ that has near optimal objective value but is far from the optimal solution $\phi$. The reason is that there can be many (equally bad) solutions for the optimal vector that recover similar objective values but are far from each other. The benefit of the dependence on $\gamma$ is that we recover an even stronger guarantee when $\mathbf{b}$ is in the span of $\mathbf{A}$. In fact, we recover the optimal solution exactly since $\gamma = 0$ in this case.

*Proof of Corollary 3.4.* We have

$$\|\mathbf{A}\hat{\phi} - \mathbf{b}\|_2^2 = \|\mathbf{A}\hat{\phi} - \mathbf{A}\phi + \mathbf{A}\phi - \mathbf{b}\|_2^2 = \|\mathbf{A}\hat{\phi} - \mathbf{A}\phi\|_2^2 + \|\mathbf{A}\phi - \mathbf{b}\|_2^2 \tag{12}$$

where the second equality follows because $\mathbf{A}\phi - \mathbf{b}$ is orthogonal to any vector in the span of $\mathbf{A}$. To see why this orthogonality holds, consider that $\phi$ minimizes $\|\mathbf{A}\phi - \mathbf{b}\|_2^2$, implying $\mathbf{A}^T(\mathbf{A}\phi - \mathbf{b}) = \mathbf{0}$. From the zero gradient condition,

$$(\mathbf{A}\hat{\phi} - \mathbf{A}\phi)^T(\mathbf{A}\phi - \mathbf{b}) = (\hat{\phi} - \phi)^T\mathbf{A}^T(\mathbf{A}\phi - \mathbf{b}) = (\hat{\phi} - \phi)^T\mathbf{0} = 0, \tag{13}$$

which confirms that $\mathbf{A}\phi - \mathbf{b}$ is orthogonal to $\mathbf{A}\hat{\phi} - \mathbf{A}\phi$.

Then by assumption, we have $\|\mathbf{A}\hat{\phi} - \mathbf{A}\phi\|_2^2 \le \epsilon\|\mathbf{A}\phi - \mathbf{b}\|_2^2$. By Observation 3.1, we have

$$\|\mathbf{A}\phi\|_2^2 = \phi^\top\mathbf{A}^\top\mathbf{A}\phi = 2^{n-2}\|\phi\|_2^2 \tag{14}$$

and, similarly, $\|\mathbf{A}(\hat{\phi} - \phi)\|_2^2 = 2^{n-2}\|\hat{\phi} - \phi\|_2^2$. Then, with the definition of $\gamma$,

$$2^{n-2}\|\hat{\phi} - \phi\|_2^2 = \|\mathbf{A}(\hat{\phi} - \phi)\|_2^2 \le \epsilon\|\mathbf{A}\phi - \mathbf{b}\|_2^2 = \epsilon\gamma\|\mathbf{A}\phi\|_2^2 = 2^{n-2}\epsilon\gamma\|\phi\|_2^2. \tag{15}$$

The statement then follows after dividing both sides by $2^{n-2}$. □

Because of the structure of $\mathbf{A}$, Theorem 3.3 and Corollary 3.4 are equivalent. Therefore, since we believe that Theorem 3.3 is nearly optimal, we believe that $\gamma$ is a necessary term in the error bound of Corollary 3.4.

Additionally, we emphasize that Kernel Banzhaf and its theoretical guarantees apply to Banzhaf values defined on any set function.

## 4 EXPERIMENTS

We conduct a detailed experimental evaluation and compare Kernel Banzhaf against state-of-the-art estimators across eight popular datasets with varying size (having between 8 and 241 features) These datasets have been used in prior research for semi-value-based model explanation (Lundberg & Lee, 2017; Covert & Lee, 2020; Lundberg et al., 2020; Karczmarz et al., 2022), and they are described in detail in Appendix C. To assess the accuracy of the estimators, we employ the $\ell_2$-norm error as our primary error metric for evaluation. While prior work used convergence ratio as a measure of effectiveness (Wang & Jia, 2023; Covert & Lee, 2020), convergence is insufficient to measure how close the estimators are to exact Banzhaf values. We use XGBoost (Chen & Guestrin, 2016) for our models in the main experiments, which makes it possible to apply the tree-based algorithm of (Karczmarz et al., 2022) to compute exact Banzhaf values. Implementation details are given in Appendix B. We also assess the effectiveness of our approach on neural network models using smaller datasets for which Equation 2 can be used to calculate exact Banzhaf values. These experiments are presented in Appendix D and their results are consistent with those for the tree models. In addition to $\ell_2$-norm, we use the objective value naturally suggested by the linear regression formulation to measure the estimation error; the results in Appendix E corroborate the superior performance of Kernel Banzhaf.

### 4.1 COMPARING BANZHAF ESTIMATORS

We compare Kernel Banzhaf with the following methods for estimating Banzhaf values:

**MC** The Monte Carlo (MC) algorithm estimates each Banzhaf value individually. Let $\mathcal{S}_i$ be the subsets sampled for player $i \in [n]$. We have that $\sum_i |\mathcal{S}_i| = m$. The MC estimate for player $i$ is given by

$$\hat{\phi}_i^{\mathrm{MC}} = \frac{1}{|\mathcal{S}_i|} \sum_{S \in \mathcal{S}_i} [v(S \cup \{i\}) - v(S)].$$

The disadvantage of this algorithm is that each sample is used only for a single player.

**MSR** The Maximum Sample Reuse (MSR) algorithm estimates all Banzhaf values simultaneously (Wang & Jia, 2023). Let $\mathcal{S}$ be the subsets sampled for all players i.e., $|\mathcal{S}| = m$. Define $\mathcal{S}_{\ni i}$ as the sampled subsets that contain player $i$ and $\mathcal{S}_{\not\ni i}$ as the sampled subsets that do not contain player $i$. The MSR estimate for player $i$ is given by

$$\hat{\phi}_i^{\mathrm{MSR}} = \frac{1}{|\mathcal{S}_{\ni i}|} \sum_{S \in \mathcal{S}_{\ni i}} v(S) - \frac{1}{|\mathcal{S}_{\not\ni i}|} \sum_{S \in \mathcal{S}_{\not\ni i}} v(S).$$

While the MSR algorithm reuses samples, it can have high variance because the magnitude of the set function $v(S)$ is generally much larger than the marginal difference between nearby values $v(S \cup \{i\}) - v(S)$.

**Kernel Banzhaf (Excluding Pairs)** This algorithm is identical to Kernel Banzhaf (Algorithm 1) but without paired sampling.

As we subsample feature sets for estimating Banzhaf value, sample size is an important concern in real-world settings. Our first experiment investigates the error of each estimator by number of samples. As shown in Figure 2, Kernel Banzhaf outperforms the other algorithms for all datasets over different sample sizes. As expected, the error for all estimators decreases as the sample size increases. Interestingly, MSR performs worse than MC when applied to datasets with a large number of features (e.g., the NHANES dataset and larger), particularly as the number of samples increases. We believe this is because the MSR algorithm has high variance which is related to the generally large magnitude of the set function. In prior work, the MSR algorithm was shown to converge faster than the MC algorithm, but faster convergence does not always imply better accuracy.

The set functions in explainable AI tasks, being based on stochastically-trained models, often exhibit variance. We investigated how the estimation errors vary with random noise added to the set function, and as shown in Figure 3, Kernel Banzhaf and its variant without paired sampling are more robust to this noise than the other algorithms, especially when the level of noise size is low. This suggests that Kernel Banzhaf is particularly well-suited for real-world applications where the set function is approximated.

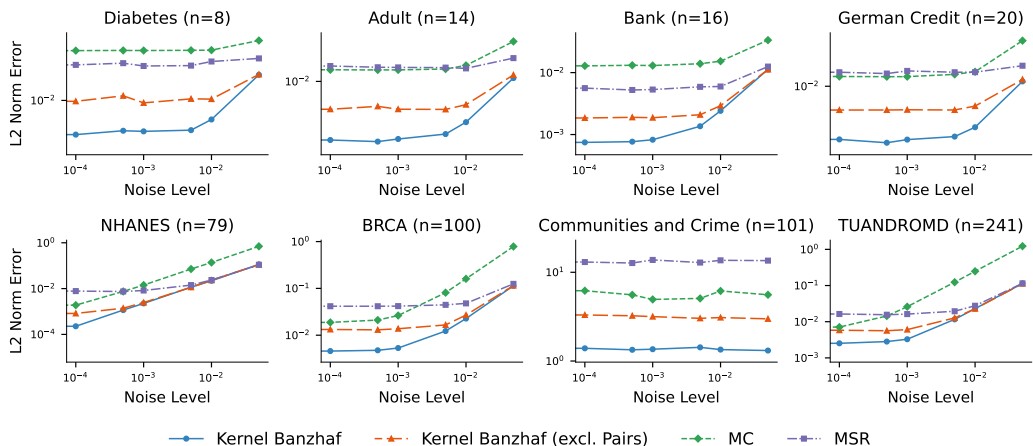

Figure 3: Plots of $\ell_2$-norm error by noise levels across Banzhaf estimators. For each noise level $\sigma$, the estimator observes $v(S) + x$ where $x \sim N(0, \sigma)$. Kernel Banzhaf outperforms for lower noise levels, eventually matching its ablated version and MSR for larger noise. MC is worse for all noise settings, likely because its constituent estimates are $v(S \cup \{i\}) + x - v(S) - x'$, essentially doubling the noise variance.

## 4.2 COMPARING KERNEL AND SHAPLEY REGRESSIONS

Prior work found that Banzhaf values can be estimated more efficiently and robustly than Shapley values (Karczmarz et al., 2022; Wang & Jia, 2023). To this end, we compare Kernel Banzhaf with state-of-the-art algorithms for estimating Shapley values: **Optimized KernelSHAP**, which employs paired sampling and sampling without replacement techniques (Lundberg & Lee, 2017; Covert & Lee, 2020) and **Leverage SHAP**, which employs leverage score sampling in addition to the afore-mentioned techniques (Musco & Witter, 2024). To obtain ground-truth Shapley values, we use the interventional setting of Tree SHAP (Lundberg et al., 2020). We employ the normalized $\ell_2$-norm squared error, defined as $\|\hat{\phi} - \phi\|_2^2 / \|\phi\|_2^2$ for comparison to ensure fairness.

When we compare the estimation error by varying the number of samples (Figure 9, Appendix F), we observe that for larger $n$, Kernel Banzhaf gives the best performance, followed by Leverage SHAP then Optimized KernelSHAP. Note that the larger the number of features, the bigger the difference in estimation error. When $n$ is small as in the Diabetes dataset, KernelSHAP and Leverage SHAP perform better because they sample without replacement.

We also investigate the impact of noise. To ensure parity between the Banzhaf and Shapley value estimators, we normalize the noise based on the raw output of the set function. The results are shown in Figure 4. The initial data points represent scenarios with no added noise, so our focus is primarily on the trends of the lines. The horizontal line representing Kernel Banzhaf, which remains unchanged as noise levels increase, underscores the robustness of Kernel Banzhaf compared to the Shapley value estimators.

Since Kernel Banzhaf is similar in design to KernelSHAP and Leverage SHAP, at first sight, it is surprising that Kernel Banzhaf performs better for large $n$. For both Kernel Banzhaf and Leverage SHAP, a crucial component of the analysis is to build the subsampled matrix $\tilde{\mathbf{A}}$ so that

$$c\mathbf{A}^\top \mathbf{A} \preceq \tilde{\mathbf{A}}^\top \tilde{\mathbf{A}} \preceq C\mathbf{A}^\top \mathbf{A} \tag{16}$$

where $c \leq C$ are two close scalars and $\preceq$ denotes the Loewner order. Multiplying on the left and right by $(\mathbf{A}^\top \mathbf{A})^{-\frac{1}{2}}$ yields

$$c\mathbf{I} \preceq (\mathbf{A}^\top \mathbf{A})^{-\frac{1}{2}} \tilde{\mathbf{A}}^\top \tilde{\mathbf{A}} (\mathbf{A}^\top \mathbf{A})^{-\frac{1}{2}} \preceq C\mathbf{I}. \tag{17}$$

Let $\mathbf{K} = (\mathbf{A}^\top \mathbf{A})^{-\frac{1}{2}} \tilde{\mathbf{A}}^\top \tilde{\mathbf{A}} (\mathbf{A}^\top \mathbf{A})^{\frac{1}{2}}$. Therefore, we can measure how close the best $C$ and $c$ are by computing the condition number of $\mathbf{K}$; a lower condition number implies $C$ is closer to $c$ and hence the subsampling preserves the important properties of the regression problem. Figure 5 plots the

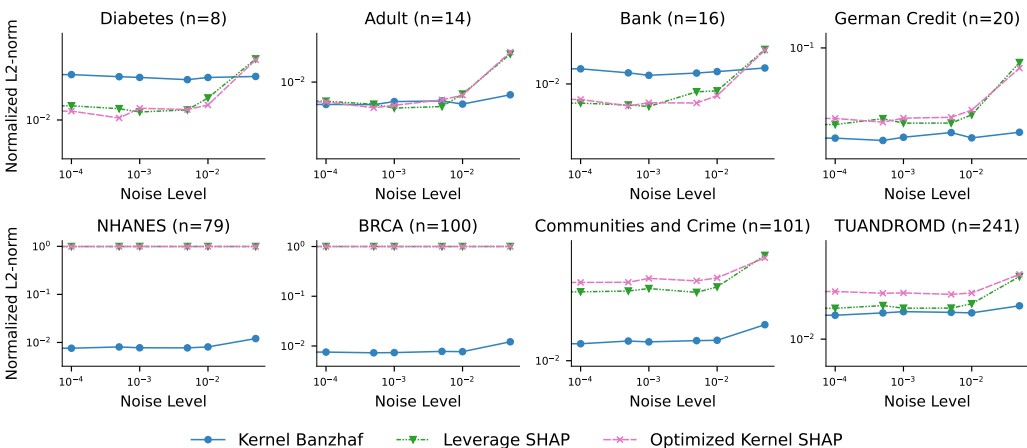

Figure 4: Normalized $\ell_2$-norm error (i.e. $\|\hat{\phi} - \phi\|_2^2/\|\phi\|_2^2$) by noise level for Banzhaf and KernelSHAP estimators. This plot displays the $\ell_2$-norm error as a function of noise level across eight datasets similar to Figure 3. Note that a more horizontal trend line is preferable. Kernel Banzhaf consistently exhibits superior performance by maintaining lower error rates amid increasing noise, compared to its competitors whose error rates rise. This demonstrates Kernel Banzhaf's robustness and efficiency, marked by its minimal loss of precision.

condition number of this matrix for all three regression-based methods; the plots suggest that Kernel Banzhaf constructs a better subproblem at all scales, explaining its superior performance.

## 5 RELATED WORK

**Game-Theoretic Explainable AI** Semivalues like Shapley and Banzhaf values, derived from game theory, are widely used for machine learning tasks. *Feature attribution* involves determining the impact of individual features on model predictions, with Shapley values notably used in Lundberg & Lee (2017). This method also facilitates *feature selection* by modifying set functions to measure the feature's impact on model performance via loss metrics (Covert et al., 2020). *Data valuation* extends to treat individual data points as players, assessing their influence on model training outcomes like accuracy or AUC (Ghorbani & Zou, 2019).

A parallel trend in utilizing Banzhaf values has been noted, with studies indicating their robustness compared to Shapley values, particularly under conditions of low numerical precision (Karczmarz et al., 2022). Furthermore, Wang & Jia (2023) established the superiority of Banzhaf values among semivalues regarding *safety margin*, the threshold of set function perturbations that the value order can withstand without changing. These properties have led to extensive application of Banzhaf values for feature attribution (Datta et al., 2015; Kulynych & Troncoso, 2017; Sliwinski et al., 2018; Patel et al., 2021; Karczmarz et al., 2022) and, more recently, in data valuation, where Wang & Jia (2023) and Li & Yu (2024) have shown their superior performance compared to Shapley values.

**Shapley Estimators** KernelSHAP, introduced by Lundberg & Lee (2017), employs game-theoretic Shapley values in a linear regression framework to explain any machine learning model's predictions. The method's efficiency and applicability are enhanced through paired sampling and sampling without replacement, which balances the number of samples used to update each Shapley value, as refined by Covert & Lee (2020); Jethani et al. (2021). In recent work, Musco & Witter (2024) proposed Leverage SHAP, a variant of KernelSHAP that uses leverage score sampling to build the subsampled linear regression problem. Leverage scores are a statistical quantity that intuitively measure the importance of a data point (Woodruff et al., 2014; Drineas & Mahoney, 2018). Leverage SHAP outperforms KernelSHAP and, because of the mathematical properties of leverage scores, provably returns accurate approximations. Meanwhile, TreeSHAP, developed by Lundberg et al. (2020), specifically tailors the SHAP framework for tree-based models, ensuring faster, exact computations by directly incorporating tree structures into Shapley value calculations.

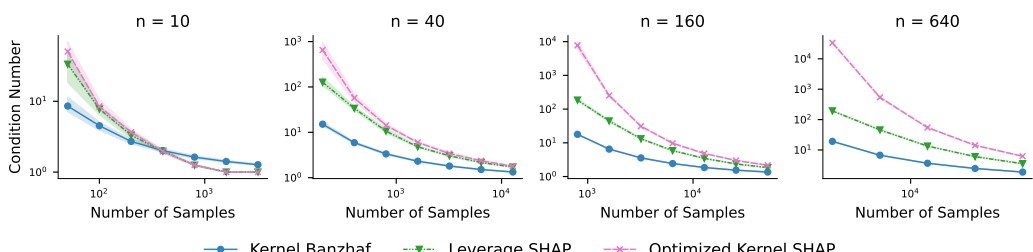

Figure 5: Plot of condition number of matrix **K** as defined in Equation 17 by sample size for estimators based on linear regression in the context of Banzhaf and Shapley value estimations. This plot demonstrates the numerical stability of the Kernel Banzhaf estimator by showing its lower condition numbers across varying sample sizes, highlighting its superior robustness and reduced sensitivity to numerical errors as shown in Figure 4, particularly when $n$ is large.

**Banzhaf Estimators**    Merrill III (1982) initially introduced approximating the Banzhaf power index through Monte Carlo sampling across subsets. Building on this, Bachrach et al. (2010) applied Hoeffding bounds to analyze the sample complexity required for these approximations. Further advancing the method, Wang & Jia (2023) developed the Maximum Sample Reuse (MSR) Monte Carlo estimator. This approach samples sets individually for each term in the marginal utility calculation, facilitating the reuse of samples across different computations of marginal utility. However, due to this variation, the MSR estimates of Banzhaf values are correlated among different players and exhibit large variance. Moreover, for structured set functions such as decision trees, where predictions are aggregated from root to leaf, Banzhaf values can be computed with exact precision (Karczmarz et al., 2022). This computational efficiency is attributable to the ability to independently isolate and assess the impact of each feature along the decision path. Despite its effectiveness within this context, this method lacks generalizability to other models and machine learning tasks.

**Banzhaf Values and Regression**    Hammer & Holzman (1992) shows how least squares regression can approximate the *Banzhaf power index*, which only requires a 'simple' set function $v : 2^N \rightarrow \{0, 1\}$. Their regression formulation differs from what we propose in Theorem 3.2 and may not yield exact Banzhaf values even when no subsampling is done. Alonso-Meijide et al. (2015) proposed modifications of Banzhaf values to enable them to satisfy the Efficiency property, via additive and multiplicative normalization, and formulated a least squares regression to obtain them.

## 6 Conclusion

In this paper, we presented Kernel Banzhaf, an innovative estimator for Banzhaf values that leverages linear regression to improve the efficiency and robustness of Banzhaf value estimation for explainable AI. Our empirical and theoretical analysis reveal that Kernel Banzhaf outperforms existing methods for Banzhaf estimation. Additionally, our findings highlight its superior robustness compared to Shapley estimators, establishing Kernel Banzhaf value as a viable and effective alternative to other semivalue-based methods for feature explanation in machine learning models.

Future research can focus on extending Kernel Banzhaf to diverse set functions and examine its potential for integration into real-time systems. This will help us to further understand the scalability and applicability of our approach in diverse machine learning contexts. Additionally, a concrete direction for future work is to develop adaptations of Kernel Banzhaf for weighted Banzhaf values (Li & Yu, 2024).

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

## A  THEORETICAL GUARANTEES

**Theorem 3.3.** *If $m = O(n \log \frac{n}{\delta} + \frac{n}{\delta \epsilon})$, Algorithm 1 produces an estimate $\hat{\phi}$ that satisfies*

$$\|\mathbf{A}\hat{\phi} - \mathbf{A}\phi\|_2 \leq (1 + \epsilon)\|\mathbf{A}\phi - \mathbf{b}\|_2 \tag{10}$$

*with probability $1 - \delta$.*

Theorem 3.3 is a standard guarantee for leverage score sampling. However, because rows are sampled in *pairs*, we need to substantially modify the standard analysis. In particular, both the spectral guarantee that the sampling matrix preserves eigenvalues and the Frobenius guarantee that the sampling matrix preserves Frobenius norm need to be reproved.

We will adopt the notation from Wu (2018); Musco & Witter (2024). Let's consider a leverage score sampling method where rows are selected in blocks. Define $\Theta$ as a partition of blocks, each containing 2 elements with identical leverage scores in our setting. We assign a sampling probability $p_i^+$ to each block $\Theta_i$, calculated as the sum of leverage scores in that block divided by the total sum of all leverage scores: $p^+i := \frac{\sum_{k \in \Theta_i \ell_k}}{\sum_j \ell_j}$. For simplicity of notation, suppose $m$ is even. Let $\mathbf{S} \in \mathbb{R}^{m \times \rho}$ represent our random sampling matrix, initially filled with zeros. The sampling process repeats $m/2$ times: Sample a block $\Theta_i$ with probability $p_i^+$. For each $k \in \Theta_i$, set the $k$th entry in an empty row to $\frac{1}{\sqrt{p_i^+}}$.

To analyze the solution obtained from this block-wise leverage score sampling, we will demonstrate that the sampling matrix $\mathbf{S}$ preserves both the spectral norm and the Frobenius norm.

**Lemma A.1** (Spectral Approximation). *Let $\mathbf{U} \in \mathbb{R}^{\rho \times n}$ be a matrix with orthonormal columns. Consider the block random sampling matrix $\mathbf{S}$ described above with rows sampled according to the leverage scores of $\mathbf{U}$. When $m = \Omega(n \log(n/\delta)/\epsilon^2)$,*

$$\|\mathbf{I} - \mathbf{U}^\top \mathbf{S}^\top \mathbf{S} \mathbf{U}\|_2 \leq \epsilon \tag{18}$$

*with probability $1 - \delta$.*

*Proof of Lemma A.1.* We will use the following matrix Chernoff bound (see e.g., Fact 1 in Woodruff et al. (2014)).

**Fact A.2** (Matrix Chernoff). *Let $\mathbf{X}_1, \ldots, \mathbf{X}_m \in \mathbb{R}^{n \times n}$ be independent samples of symmetric random matrices with $\mathbb{E}[\mathbf{X}_j] = \mathbf{0}$, $\|\mathbf{X}_j\|_2 \leq \gamma$ for all $j$, and $\|\mathbb{E}_j[\mathbf{X}_j^2]\|_2 \leq \sigma^2$. Then for any $\epsilon > 0$,*

$$\Pr\left(\left\|\frac{1}{m}\sum_{j=1}^m \mathbf{X}_j\right\|_2 \geq \epsilon\right) \leq 2n \exp\left(\frac{-m\epsilon^2}{\sigma^2 + \gamma\epsilon/3}\right). \tag{19}$$

For sample $j \in [m]$, let $i(j)$ be the index of the block selected. Define

$$\mathbf{X}_j = \mathbf{I} - \frac{1}{p_{i(j)}^+} \sum_{k \in \Theta_{i(j)}} \mathbf{U}_k^\top \mathbf{U}_k \tag{20}$$

We will compute $\mathbf{E}[X_j]$, $\|\mathbf{X}_j\|_2$, and $\|\mathbb{E}[\mathbf{X}_j^2]\|_2$. First,

$$\mathbb{E}[\mathbf{X}_j] = \mathbf{I} - \sum_{i=1}^{|\Theta|} p_i^+ \frac{1}{p_i^+} \sum_{k \in \Theta_i} \mathbf{U}_k^\top \mathbf{U}_k = \mathbf{0} \tag{21}$$

where the last equality follows because $\Theta$ is a partition and $\mathbf{U}^\top \mathbf{U} = \mathbf{I}$. Next, note that

$$\|\mathbf{X}_j\|_2 \leq \|\mathbf{I}\|_2 + \frac{\sum_{k \in \Theta_{i(j)}} \|\mathbf{U}_k^\top \mathbf{U}_k\|_2}{\sum_{k \in \Theta_{i(j)}} p_k} = 1 + n\frac{\sum_{k \in \Theta_{i(j)}} \|\mathbf{U}_k\|_2^2}{\sum_{k \in \Theta_{i(j)}} \ell_k} = 1 + n \tag{22}$$

where the last equality follows because $\|\mathbf{U}_k\|_2^2 = \ell_k$ since $\mathbf{U}^\top\mathbf{U} = \mathbf{I}$. Define $\mathbf{U}_{\Theta_i} \in \mathbb{R}^{|\Theta_i|\times n}$ as the matrix with rows $\mathbf{U}_k$ for $k \in \Theta_i$. Observe that $\sum_{k\in\Theta_i}\mathbf{U}_k^\top\mathbf{U}_k = \mathbf{U}_{\Theta_i}^\top\mathbf{U}_{\Theta_i}$. Finally, note that

$$\mathbb{E}[\mathbf{X}_j^2] = \mathbf{I} - 2\sum_{i=1}^{|\Theta|}p_i^+\frac{\sum_{k\in\Theta_i}\mathbf{U}_k^\top\mathbf{U}_k}{p_i^+} + \sum_{i=1}^{|\Theta|}p_i^+\frac{\left(\mathbf{U}_{\Theta_i}^\top\mathbf{U}_{\Theta_i}\right)^2}{p_i^{+2}} \tag{23}$$

$$= -\mathbf{I} + \sum_{i=1}^{|\Theta|}\frac{1}{p_i^+}\mathbf{U}_{\Theta_i}^\top\mathbf{U}_{\Theta_i}\mathbf{U}_{\Theta_i}^\top\mathbf{U}_{\Theta_i}. \tag{24}$$

Observe that entry $(k,k')$ of $\mathbf{U}_{\Theta_i}\mathbf{U}_{\Theta_i}^\top \in \mathbb{R}^{|\Theta_i|\times|\Theta_i|}$ is $\mathbf{U}_k\mathbf{U}_{k'}^\top$. So the absolute value of each entry is $|\mathbf{U}_k\mathbf{U}_{k'}^\top| \le \|\mathbf{U}_k\|_2\|\mathbf{U}_{k'}\|_2 = \ell_k^{1/2}\ell_{k'}^{1/2}$ by Cauchy-Schwarz. Define $\ell_i^{\max} = \max_{k\in\Theta_i}\ell_k$ and $\ell_i^{\min} = \min_{k\in\Theta_i}\ell_k$. By the Gershgorin circle theorem, $\mathbf{U}_{\Theta_i}\mathbf{U}_{\Theta_i}^\top \preceq 2\ell_i^{\max}|\Theta_i|^2\mathbf{I}$. Equivalently, $\mathbf{A} \preceq \mathbf{B}$, $\mathbf{x}^\top\mathbf{A}\mathbf{x} \le \mathbf{x}^\top\mathbf{B}\mathbf{x}$ for all $\mathbf{x}$. Consider an arbitrary $\mathbf{z}$. We have $\mathbf{z}^\top\mathbf{C}^\top\mathbf{A}\mathbf{C}\mathbf{z} \le \mathbf{z}^\top\mathbf{C}^\top\mathbf{B}\mathbf{C}\mathbf{z}$ since $\mathbf{C}\mathbf{z}$ is some $\mathbf{x}$. It follows that $\mathbf{U}_{\Theta_i}^\top\mathbf{U}_{\Theta_i}\mathbf{U}_{\Theta_i}^\top\mathbf{U}_{\Theta_i} \preceq 2\ell_i^{\max}|\Theta_i|^2\mathbf{U}_{\Theta_i}^\top\mathbf{U}_{\Theta_i}$. Then

$$(24) \preceq n\sum_{i=1}^{|\Theta_i|}\frac{2\ell_i^{\max}|\Theta_i|^2\mathbf{U}_{\Theta_i}^\top\mathbf{U}_{\Theta_i}}{\sum_{k\in\Theta_i}\ell_k} \preceq n\max_i 2|\Theta_i|\frac{\ell_i^{\max}}{\ell_i^{\min}}\mathbf{I}. \tag{25}$$

Since $|\Theta_i| \le 2$ and the leverage scores in a block are all equal, $\|\mathbb{E}[\mathbf{X}_j^2]\|_2 \le 4n$.

Applying Fact A.2 with $m = O(n\log(n/\delta)/\epsilon^2)$ yields

$$\Pr\left(\left\|\frac{1}{m}\sum_{j=1}^m\mathbf{I} - \frac{1}{p_{i(j)}^+}\mathbf{U}_{\Theta_i}^\top\mathbf{U}_{\Theta_i}\right\|_2 \ge \epsilon\right) \le \delta. \tag{26}$$

The lemma statement follows.

$\square$

We will also show that the sampling matrix preserves the Frobenius norm.

**Lemma A.3** (Frobenius Approximation). *Consider the block random sampling matrix $\mathbf{S}$ described above with rows sampled according to the leverage scores of $\mathbf{U} \in \mathbb{R}^{\rho\times n}$. Let $\mathbf{V} \in \mathbb{R}^{\rho\times n'}$. As long as $m \ge \frac{1}{\delta\epsilon^2}$, then*

$$\left\|\mathbf{U}^\top\mathbf{S}^\top\mathbf{S}\mathbf{V} - \mathbf{U}^\top\mathbf{V}\right\|_F \le \epsilon\|\mathbf{U}\|_F\|\mathbf{V}\|_F \tag{27}$$

*with probability $1 - \delta$.*

*Proof.* By Proposition 2.2 in Wu (2018), we have that

$$\mathbb{E}[\|\mathbf{U}^\top\mathbf{S}^\top\mathbf{S}\mathbf{V} - \mathbf{U}^\top\mathbf{V}\|_F^2] \le \frac{1}{m}\sum_{j=1}^{|\Theta|}\frac{1}{p_{i(j)}^+}\|\mathbf{U}_{\Theta_{i(j)}}\|_F^2\|\mathbf{V}_{\Theta_{i(j)}}\|_F^2 \tag{28}$$

where $\mathbf{U}_{\Theta_i} \in \mathbb{R}^{n\times|\Theta_i|}$ is a matrix with columns $\mathbf{U}_k$ for $k \in \Theta_i$. Because $p_k = \|\mathbf{U}_k\|_2^2/\|\mathbf{U}\|_F^2$ by the definition of leverage scores, we have

$$\mathbb{E}[\|\mathbf{U}^\top\mathbf{S}^\top\mathbf{S}\mathbf{V} - \mathbf{U}^\top\mathbf{V}\|_F^2] \le \frac{1}{m}\sum_{i=1}^{|\Theta|}\|\mathbf{U}\|_F^2\frac{\sum_{k\in\Theta_i}\|\mathbf{U}_k\|_2^2}{\sum_{k\in\Theta_i}\|\mathbf{U}_k\|_2^2}\|\mathbf{V}_{\Theta_i}\|_F^2 = \frac{1}{m}\|\mathbf{U}\|_F^2\|\mathbf{V}\|_F^2. \tag{29}$$

By Markov's inequality,

$$\Pr\left(\left\|\mathbf{U}^\top\mathbf{S}^\top\mathbf{S}\mathbf{V} - \mathbf{U}^\top\mathbf{V}\right\|_F > \epsilon\|\mathbf{U}\|_F\|\mathbf{V}\|_F\right) \le \frac{\mathbb{E}\left[\left\|\mathbf{U}^\top\mathbf{S}^\top\mathbf{S}\mathbf{V} - \mathbf{U}^\top\mathbf{V}\right\|_F^2\right]}{\epsilon^2\|\mathbf{U}\|_F^2\|\mathbf{V}\|_F^2} \le \frac{1}{m\epsilon^2} \le \delta \tag{30}$$

as long as $m \ge \frac{1}{\delta\epsilon^2}$. $\square$

With Lemmas A.1 and A.3 already proved for the special paired leverage score sampling, the following analysis is standard. For example, see Appendix A in Musco & Witter (2024) which proves the same guarantee but for sampling *without* replacement, requiring yet different proofs of the spectral and Frobenius approximation lemmas. We include the following proof for completeness.

*Proof of Theorem 3.3.* Observe that

$$\|\mathbf{A}\tilde{\phi} - \mathbf{b}\|_2^2 = \|\mathbf{A}\tilde{\phi} - \mathbf{A}\phi + \mathbf{A}\phi - \mathbf{b}\|_2^2 = \|\mathbf{A}\tilde{\phi} - \mathbf{A}\phi\|_2^2 + \|\mathbf{A}\phi - \mathbf{b}\|_2^2 \tag{31}$$

where the second equality follows because $\mathbf{A}\phi - \mathbf{b}$ is orthogonal to any vector in the span of $\mathbf{A}$. So to prove the theorem, it suffices to show that

$$\|\mathbf{A}\tilde{\phi} - \mathbf{A}\phi\|_2^2 \le \epsilon\|\mathbf{A}\phi - \mathbf{b}\|_2^2. \tag{32}$$

Let $\mathbf{U} \in \mathbb{R}^{\rho \times n}$ be an orthonormal matrix that spans the columns of $\mathbf{A}$. There is some $\mathbf{y}$ such that $\mathbf{U}\mathbf{y} = \mathbf{A}\phi$ and some $\tilde{\mathbf{y}}$ such that $\mathbf{U}\tilde{\mathbf{y}} = \mathbf{A}\tilde{\phi}$. Observe that $\|\mathbf{A}\tilde{\phi} - \mathbf{A}\phi\|_2 = \|\mathbf{U}\tilde{\mathbf{y}} - \mathbf{U}\mathbf{y}\|_2 = \|\tilde{\mathbf{y}} - \mathbf{y}\|_2$ where the last equality follows because $\mathbf{U}^\top\mathbf{U} = \mathbf{I}$.

By the reverse triangle inequality and the submultiplicavity of the spectral norm, we have

$$\|\tilde{\mathbf{y}} - \mathbf{y}\|_2 \le \|\mathbf{U}^\top\mathbf{S}^\top\mathbf{S}\mathbf{U}(\tilde{\mathbf{y}} - \mathbf{y})\|_2 + \|\mathbf{U}^\top\mathbf{S}^\top\mathbf{S}\mathbf{U}(\tilde{\mathbf{y}} - \mathbf{y}) - (\tilde{\mathbf{y}} - \mathbf{y})\|_2 \tag{33}$$

$$\le \|\mathbf{U}^\top\mathbf{S}^\top\mathbf{S}\mathbf{U}(\tilde{\mathbf{y}} - \mathbf{y})\|_2 + \|\mathbf{U}^\top\mathbf{S}^\top\mathbf{S}\mathbf{U} - \mathbf{I}\|_2\|\tilde{\mathbf{y}} - \mathbf{y}\|_2. \tag{34}$$

Because $\mathbf{U}$ has the same leverage scores as $\mathbf{A}$ and the number of rows sampled in $\mathbf{S}$ is within a constant factor of $m$, we can apply Lemma A.1: With $m = O(n \log\frac{n}{\delta})$, we have $\|\mathbf{U}^\top\mathbf{S}^\top\mathbf{S}\mathbf{U} - \mathbf{I}\|_2 \le \frac{1}{2}$ with probability $1 - \delta/2$. So, with probability $1 - \delta/2$,

$$\|\tilde{\mathbf{y}} - \mathbf{y}\|_2 \le 2\|\mathbf{U}^\top\mathbf{S}^\top\mathbf{S}\mathbf{U}(\tilde{\mathbf{y}} - \mathbf{y})\|_2. \tag{35}$$

Then

$$\|\mathbf{U}^\top\mathbf{S}^\top\mathbf{S}\mathbf{U}(\tilde{\mathbf{y}} - \mathbf{y})\|_2 = \left\|\mathbf{U}^\top\mathbf{S}^\top\left(\mathbf{S}\mathbf{U}\tilde{\mathbf{y}} - \mathbf{S}\mathbf{b} + \mathbf{S}\mathbf{b} - \mathbf{S}\mathbf{U}\mathbf{y}\right)\right\|_2 \tag{36}$$

$$= \left\|\mathbf{U}^\top\mathbf{S}^\top\mathbf{S}\left(\mathbf{U}\mathbf{y} - \mathbf{b}\right)\right\|_2 \tag{37}$$

where the second equality follows because $\mathbf{S}\mathbf{U}\tilde{\mathbf{y}} - \mathbf{S}\mathbf{b}$ is orthogonal to any vector in the span of $\mathbf{S}\mathbf{U}$. By similar reasoning, notice that $\mathbf{U}^\top(\mathbf{U}\mathbf{y} - \mathbf{b}) = \mathbf{0}$. Then, as long as $m = O(\frac{n}{\delta\epsilon})$, we have

$$\left\|\mathbf{U}^\top\mathbf{S}^\top\mathbf{S}\left(\mathbf{U}\mathbf{y} - \mathbf{b}\right)\right\|_2 \le \frac{\sqrt{\epsilon}}{2\sqrt{n}}\|\mathbf{U}\|_F\|\mathbf{U}\mathbf{y} - \mathbf{b}\|_2 \tag{38}$$

with probability $1 - \delta/2$ by Lemma A.3. Since $\mathbf{U}$ has orthonormal columns, $\|\mathbf{U}\|_F^2 \le n$. Then, combining inequalities yields

$$\|\mathbf{A}\tilde{\phi} - \mathbf{A}\phi\|_2^2 = \|\tilde{\mathbf{y}} - \mathbf{y}\|_2^2 \le 2\|\mathbf{U}^\top\mathbf{S}^\top\mathbf{S}\mathbf{U}(\tilde{\mathbf{y}} - \mathbf{y})\|_2 \le \epsilon\|\mathbf{U}\mathbf{y} - \mathbf{b}\|_2 = \epsilon\|\mathbf{A}\phi - \mathbf{b}\|_2^2 \tag{39}$$

with probability $1 - \delta$. $\qquad\square$

## B  Implementation Details

**Feature perturbation and exact Banzhaf value calculation**  There are generally two approaches to handling removed features in feature perturbation for general set functions, as discussed in Chen et al. (2020) and Kumar et al. (2020). Given an explicand $x$ and a subset of features $S$, define $\mathbf{x}^S$ as the observation where $\mathbf{x}_i^S = \mathbf{x}_i^e$ if feature $i \in S$ and, otherwise, $\mathbf{x}^{\bar{S}}$ is sampled from a distribution. The first method involves sampling from the conditional distribution of the removed features, where replacement values for the absent features are sampled according to $\mathbf{x}^{\bar{S}} \sim p(\mathbf{x}^{\bar{S}}|\mathbf{x}^S)$. This approach, while precise, is computationally expensive. Alternatively, the marginal distribution can be used where the observed features $\mathbf{x}^S$ are ignored, and replacement values are sampled according to $\mathbf{x}^{\bar{S}} \sim p(\mathbf{x}^{\bar{S}})$. Due to its lower computational complexity, we adopt the latter approach.

For implementation, distinct feature perturbation methods are applied depending on the set function, i.e. model type. For *tree-based models*, as we need to use the tree-based algorithm for calculating the ground truth Banzhaf values to measure errors, we utilize a method aligned with Algorithm 1 from Karczmarz et al. (2022), which computes predictions using partial features. Specifically, during tree traversal, if feature $i \in S$, we proceed according to the threshold to select the child node; if $i \notin S$, we traverse both children and compute a weighted average of the predictions, effectively nullifying the influence of features not in $S$ without any feature value replacement.

For *neural network models*, instead of using fixed baseline values for the removed features, we compute the average of the model's predictions using replacement values randomly sampled from 50 baseline points, different from the explicand. The explicand $x^e$ is repeated 50 times, with the non-selected features replaced by values from these baseline points, and the average of $M(x^S)$ is taken to estimate the impact of marginalizing out the non-selected features. To calculate ground truth Banzhaf values, we evaluate all $2^n$ subsets of features in this way.

**Sample size for Monte Carlo**  Given a sample size of $n$, the MSR and Kernel Banzhaf estimators effectively use $n$ samples to estimate Banzhaf values for all features concurrently. In contrast, the Monte Carlo method estimates each feature independently and requires a fair allocation of the total $m$ samples. To achieve this, we implement a loop where for each iteration $i = 1, \ldots, m$, we use $i \mod n + 1$ samples when $i$ is divisible by $n$, and $i \mod n$ samples otherwise. This approach ensures that the Monte Carlo method also utilizes exactly $m$ samples, maintaining consistency in sample usage across different estimators.

## C  Datasets and Models

The Diabetes dataset (Bache & Lichman, 2013), sourced from the National Institute of Diabetes and Digestive and Kidney Diseases, comprises 8 features. Its primary objective is to predict, based on diagnostic measurements, whether a patient has diabetes, thus it's categorized as a classification task. The Census Income dataset (Bache & Lichman, 2013; Covert & Lee, 2020), also known as the Adult dataset, involves predicting whether an individual's income exceeds $50K/yr based on census data, using 14 features. The Portuguese Bank Marketing dataset (Moro et al., 2014) is another classification task with 16 features aimed at predicting client subscription to a term deposit. The German Credit dataset (Bache & Lichman, 2013), known as Statlog, involves classifying individuals as having good or bad credit risks based on 20 attributes. The NHANES dataset, with 79 features derived from the National Health and Nutrition Examination Survey (NHANES) I Epidemiologic Followup Study, models the risk of death over a 20-year follow-up period, as discussed in (Lundberg et al., 2020; Karczmarz et al., 2022). For the Breast Cancer (BRCA) subtype classification dataset, 100 out of 17,814 genes were selected to minimize overfitting in a relatively small dataset of 510 patients, following guidelines from (Covert & Lee, 2020). The Communities and Crime Unnormalized dataset (Bache & Lichman, 2013) aims to predict the total number of violent crimes per 100,000 population, comprising a predictive regression task with 101 features. The Tezpur University Android Malware Dataset (TUANDROMD) (Bache & Lichman, 2013) includes 241 attributes, with the primary classification target distinguishing between malware and goodware.

These datasets vary in size and column types and are predominantly utilized in previous studies for semi-value-based model explanation (Lundberg & Lee, 2017; Covert & Lee, 2020; Lundberg et al., 2020; Karczmarz et al., 2022). We primarily focus on tabular datasets because they are more

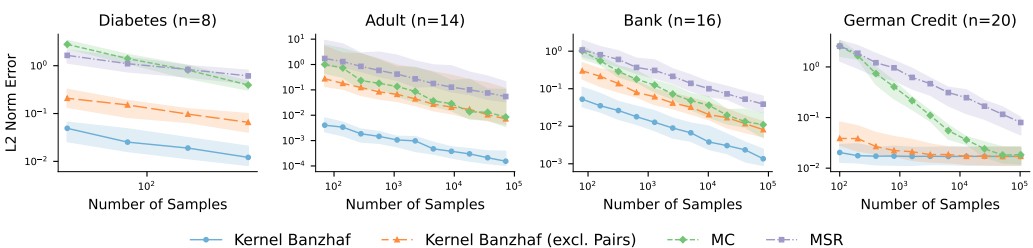

Figure 6: $\ell_2$-norm error by sample size for Banzhaf estimators in explaining neural network models. This figure compares the $\ell_2$-norm errors of Kernel Banzhaf (including an ablated version without paired sampling), MC, and MSR across increasing sample sizes on four small datasets. The results highlight the robust performance and generalizability of Kernel Banzhaf across various model types.

thoroughly studied in this field and allow for easier acquisition of ground truth, especially in large datasets, using tree-based algorithms. Additionally, tabular datasets are prevalent in scenarios involving smaller datasets with fewer features.

For the experiments involving tree set functions, we trained an XGBoost regressor model (Chen & Guestrin, 2016) with 100 trees and a maximum depth of 4. For the non-tree model experiments, we utilized a two-layer neural network equipped with a dropout layer with a rate of 0.5 to mitigate overfitting. This network was trained using a batch size of 32 and a learning rate of 0.0001, across 100 epochs. We chose this relatively simple model architecture because our primary focus is on explaining model behavior rather than maximizing its predictive accuracy.

## D  NEURAL NETWORK-BASED MODEL EXPLANATIONS

In this section, we evaluate Banzhaf estimators for explaining neural network models, where the output of the set function is the neural network's raw prediction. Calculating ground truth Banzhaf values for datasets with more than 50 features presents challenges, primarily because the tree-based algorithm for Banzhaf value calculation, as proposed in Karczmarz et al. (2022), is limited to tree models. Consequently, our experiments are confined to four smaller datasets. As illustrated in Figure 6, Kernel Banzhaf estimators, both with and without paired sampling, consistently outperform Monte Carlo (MC) and Maximum Sample Reuse (MSR) estimators in non-tree set functions. These experiments further underscore the generalizability of our algorithm across different model types.

## E  COMPLEMENTARY BANZHAF ESTIMATOR EXPERIMENTS

**Relative Objective Error**   To further demonstrate the superior performance of Kernel Banzhaf, we evaluated the Banzhaf estimators against exact Banzhaf values using relative objective error. Building on the framework where Banzhaf values solve a linear regression problem (as established in Theorem 3.2), we aim to minimize the objective function $\|\mathbf{A}\mathbf{x} - \mathbf{b}\|_2$. Here, $\phi$ represents the vector of exact Banzhaf values for each feature, and $\hat{\phi}$ represents the estimations from different estimators. We evaluate the differences between $\|\mathbf{A}\phi - \mathbf{b}\|_2$ and $\|\mathbf{A}\hat{\phi} - \mathbf{b}\|_2$, because although the optimal linear regression solution corresponds to the Banzhaf values, the optimal objective error $\|\mathbf{A}\phi - \mathbf{b}\|_2$ is not necessarily zero. Thus, we assess the relative objective error. The comparative analysis, shown in Figure 7, reveals that the plots of this error with increasing sample size and noise level are analogous to those using $\ell_2$-norm error, where Kernel Banzhaf consistently surpasses all other estimators in both efficacy and robustness. These experiments underscore the effectiveness of our Kernel Banzhaf algorithm in accurately estimating Banzhaf values and solving the corresponding linear regression challenge.

**Time Complexity**   We further evaluated the computational efficiency of the MC, MSR, and Kernel Banzhaf estimators by measuring the exact time required to estimate Banzhaf values, as depicted in

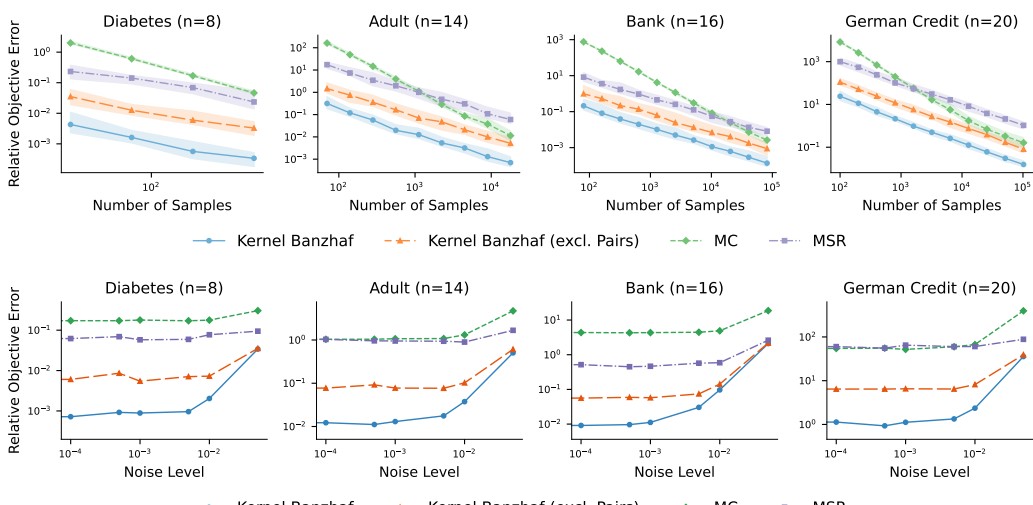

Figure 7: Relative objective error based on $\|\mathbf{A}\mathbf{x} - \mathbf{b}\|_2$ plotted against sample size (top row) and noise level (bottom row) for comparing Banzhaf estimators across four small datasets. Kernel Banzhaf consistently outperforms other estimators, demonstrating enhanced accuracy and robustness as sample sizes and noise levels vary. This performance highlights its efficacy in solving the linear regression problem where the solutions represent Banzhaf values.

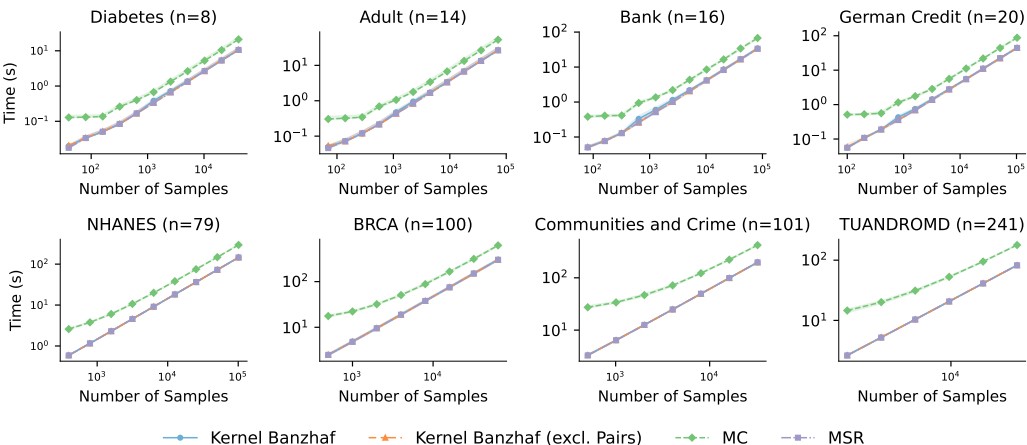

Figure 8: Computational time required for Banzhaf value estimation across varying sample sizes for eight datasets. Notably, Kernel Banzhaf (both with and without paired sampling) and MSR demonstrate comparable computational efficiencies, while MC consistently requires approximately twice the computation time, particularly evident when sample size is small.

Figure 8. All experiments were conducted on a Lenovo SD650 with 128 GB of RAM, using only one thread for computation. Our analysis reveals that both MSR and Kernel Banzhaf (with and without paired sampling) exhibit comparable computational times across all datasets and sample sizes. In contrast, the MC method consistently requires approximately twice the time of the other estimators. This discrepancy arises primarily because the most time-consuming operation involves computing the set function output $v(S)$ for a given sample $S$ of players. Kernel Banzhaf and MSR efficiently leverage a single calculation of $v(S)$ per sample for all players. In contrast, MC needs to compute

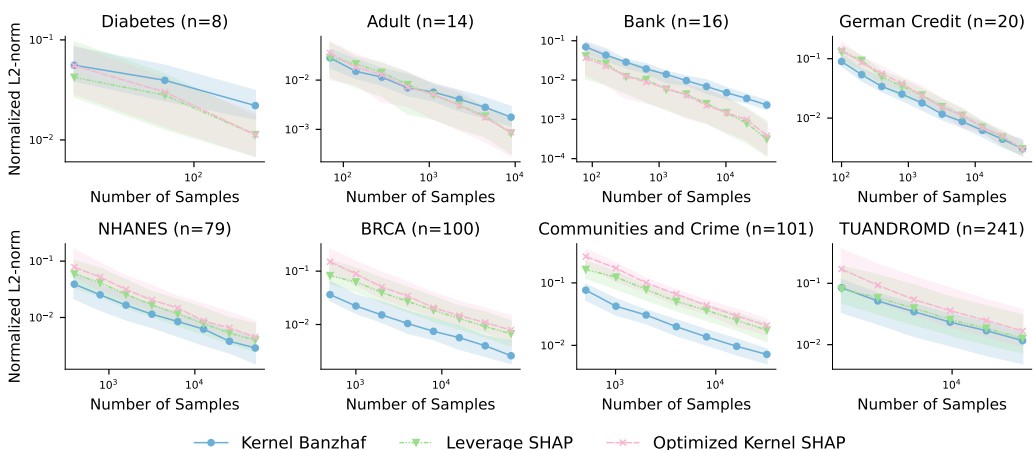

Figure 9: Normalized $\ell_2$-norm error (i.e. $\|\hat{\phi} - \phi\|_2^2/\|\phi\|_2^2$) by sample size for Kernel Banzhaf and KernelSHAP estimators. This set of plots compares the $\ell_2$-norm error across increasing sample sizes for Kernel Banzhaf, Leverage SHAP, and Optimized KernelSHAP across eight datasets. The plots aim to evaluate the efficiency and accuracy of Kernel Banzhaf (the best Banzhaf estimator) against advanced KernelSHAP methods under varying sample sizes. It demonstrates that Kernel Banzhaf outperforms KernelSHAP algorithms for larger feature sets.

both $v(S \cup i)$ and $v(S)$ to determine the marginal contribution for each feature $i$ individually, thus doubling the computation time compared to the other estimators.

## F    COMPLEMENTARY KERNEL BANZHAF VS. KERNELSHAP EXPERIMENTS

In this section, we assess the performance of Kernel Banzhaf against state-of-the-art KernelSHAP algorithms by varying sample sizes. Kernel Banzhaf, as shown in Figure 9, demonstrates superior performance on larger datasets (bottom row). For smaller datasets, the three algorithms appear comparable at first glance, since KernelSHAP are using sampling without replacement to increase the diversity of the samples; however, the smaller shaded areas for Kernel Banzhaf, representing the 25% and 75% percentiles, indicate significantly greater stability compared to KernelSHAP algorithms, which highlight Kernel Banzhaf's enhanced stability.

## G    KERNEL BANZHAF WITH SAMPLING WITHOUT REPLACEMENT

Contrary to the approach in Musco & Witter (2024), which employs sampling without replacement to enhance KernelSHAP, our empirical results, as illustrated in Figure 10, indicate that this method does not yield improvements. We observe a significant reduction in error when the sample size $m$ exceeds $2^n$, where $n$ is the number of features, in the first three small datasets. This reduction can be attributed to the fact that the algorithm has exhaustively sampled all possible subsets once $m > 2^n$. However, it is impractical to set $m$ beyond the total number of subset combinations $2^n$ in typical scenarios, as doing so would allow a complete enumeration of subsets, thereby directly computing the ground truth Banzhaf values.

In standard cases where $m < 2^n$, our sampling with replacement approach not only matches the performance of sampling without replacement but also surpasses it in terms of efficiency and scalability. This is evident as sampling using the binary representation of random integers used in sampling without replacement becomes infeasible for large $n$. Additionally, sampling with replacement ensures the collection of exactly $m$ distinct samples, in contrast to the approximate number $m'$ obtained without replacement. Given these advantages, we continue to utilize the sampling without replacement setting in our proposed algorithm, offering a comprehensive analysis that diverges from the findings presented in Musco & Witter (2024).

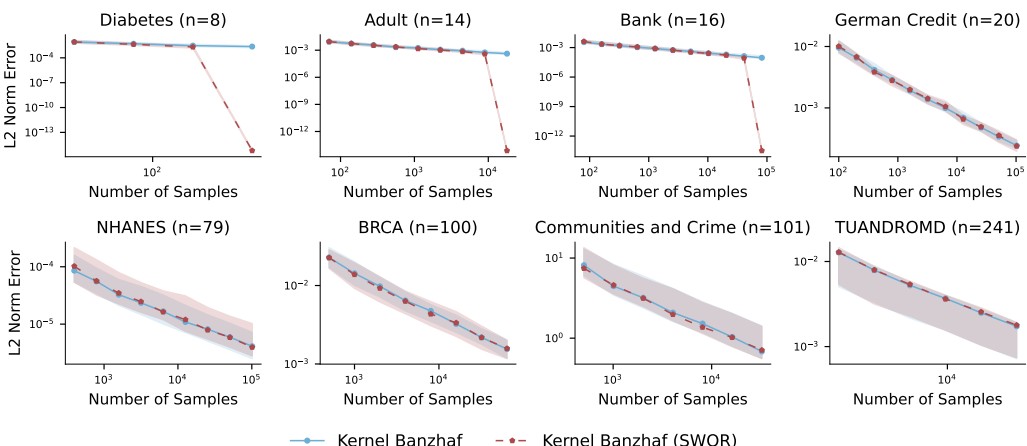

Figure 10: Comparison of $\ell_2$-norm error between Kernel Banzhaf and Kernel Banzhaf with sampling without replacement (SWOR) across eight datasets. When sample size $m < 2^n$, the performance of the two algorithms is quite similar, as illustrated by the overlapping lines. Note that the sharp error reduction in the first three small datasets occurs when sample size $m$ exceeds $2^n$, due to the SWOR algorithm exhaustively sampling all possible subsets at $m > 2^n$.

Theoretically, our sampling with replacement approach is better in terms of efficiency and its use of exact $m$ samples.

## H  PROPERTIES OF SHAPLEY VALUES AND BANZHAF VALUES

Shapley values satisfy four desirable properties: *Null Player* ensures that a player who does not contribute to any coalition, meaning their inclusion in any subset of players does not affect the overall outcome, is assigned a value of zero. *Symmetry* requires that two players who contribute equally to all possible coalitions receive the same value. *Linearity* requires that the Shapley value of a player in a combined game (formed by adding two games together) is equal to the sum of that player's Shapley values in the two individual games. *Efficiency* requires that the total value assigned to all players must sum to the value generated by the full set of players. (Shapley, 1953).

Instead of the *Efficiency* property, the Banzhaf index satisfies *2-Efficiency*, which requires that the sum of the values of any two players equals the value of these two players when considered jointly in a reduced game setting (Banzhaf, 1965; Lehrer, 1988). The necessity of the efficiency property has been debated in the context of machine learning. Sundararajan et al. (2017) suggest that the *Efficiency* property is only essential in contexts where semi-values, such as those in voting games, are interpreted numerically; Kwon & Zou (2022) argues that the utility function in machine learning applications often does not correspond directly to monetary value, so aligning the sum of data values with total utility is unnecessary. In applications where the primary goal involves ranking features according to their importance or evaluating data, the exact numerical contribution of each feature is less critical. Both Banzhaf and Shapley values, despite their theoretical disparities, often yield the same ordering of players as shown in Karczmarz et al. (2022), which suffices for these applications. Therefore, given their efficiency and robustness properties, Banzhaf values serve as particularly effective tools in machine learning tasks (Wang & Jia, 2023).

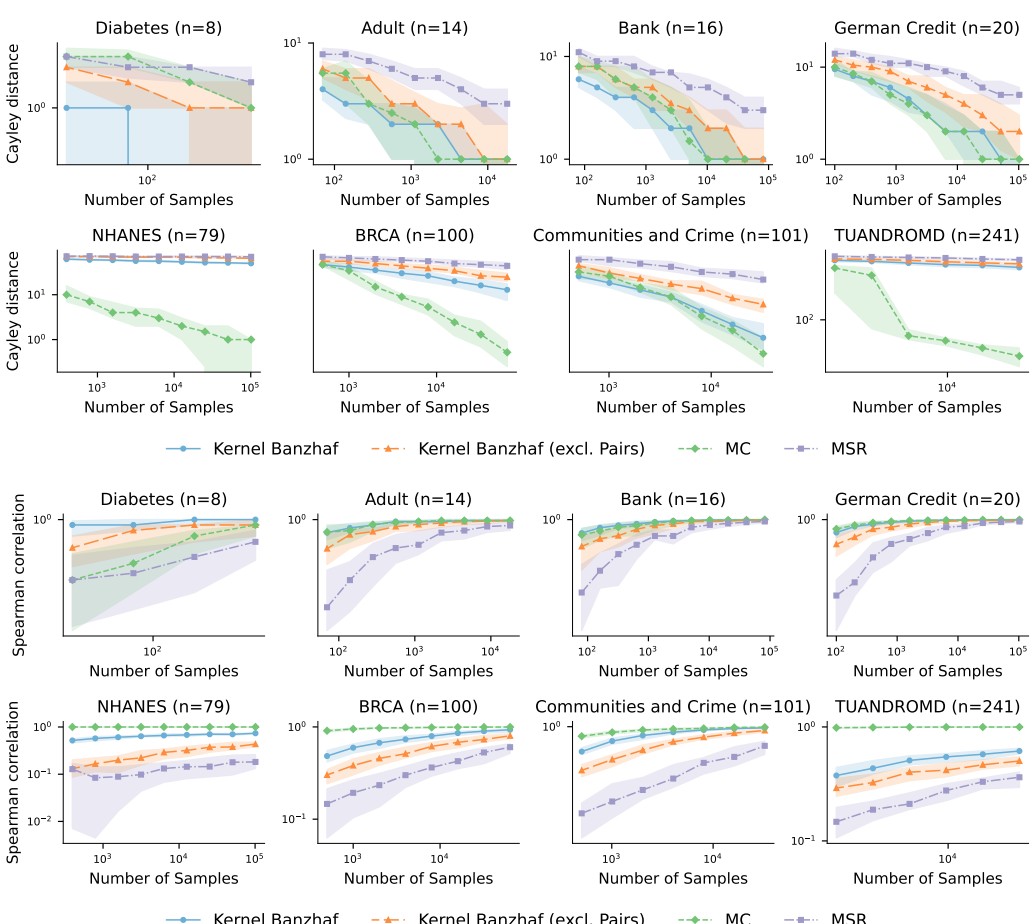

Figure 11: Comparison of feature ranking recovery Using Cayley Distance (top panel) and Spearman Correlation Coefficients (bottom panel). This figure illustrates the performance of different estimators in recovering overall feature rankings across multiple datasets. Lower Cayley distances and higher Spearman correlations indicate more accurate feature ranking recovery. Our results indicate that while the Kernel Banzhaf algorithm and the MC method perform comparably in datasets with a smaller number of features, the MC method excels as feature size increases due to its precision in assigning zero values to non-contributory features. However, the practicality of this metric is limited.

## I  EVALUATION OF BANZHAF ESTIMATORS IN FEATURE RANKING RECOVERY

Aside from evaluating the quantitative errors between estimated Banzhaf values and exact Banzhaf values, another critical metric that reveals the meaningfulness of the estimated results is how well the estimator recovers feature ranking. Feature ranking is important to feature comparison and selection, which are useful for enhancing the performance of machine learning models. Accurate feature ranking helps in identifying the most influential features, thereby facilitating more efficient and effective feature engineering and dimensionality reduction strategies.

In order to evaluate this property, we incorporate two well-known metrics: *Cayley distance* and *Spearman rank correlation*. The Cayley distance refers to the minimum number of transpositions required to transform one permutation into another. This metric provides a concrete measure of the difference between two rankings, capturing the minimal edit sequence needed, which is particularly useful in understanding the stability and reliability of feature ranking methods, and it's also adopted

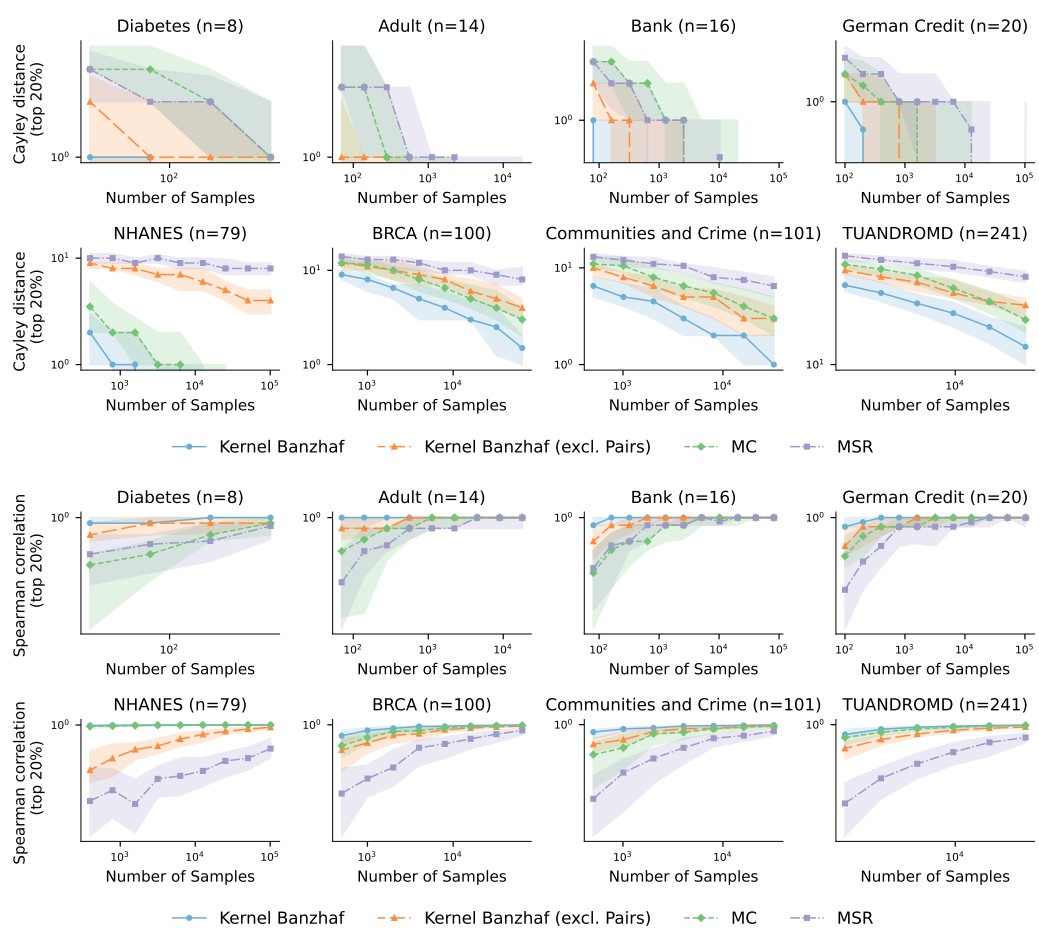

Figure 12: **Comparison of top 20% feature ranking recovery Using Cayley Distance (top panel) and Spearman Correlation Coefficients (bottom panel).** This figure illustrates the performance of four estimators in accurately recovering the rankings of the top 20% most influential features, or the top 7 features when $n \leq 20$), across multiple datasets. The plots show that our Kernel Banzhaf algorithm consistently matches or outperforms the MC method, confirming its effectiveness in the estimation of the most significant features. This metric is crucial in real-world applications as it prioritizes top features, which is particularly valuable in large datasets where distinguishing among lower-ranked features can be challenging.

in Karczmarz et al. (2022). Spearman's rank correlation, $\rho$, on the other hand, measures the strength and direction of association between two ranked variables. Formally, it is defined as the Pearson correlation coefficient between the rank values of the variables, mathematically expressed as:

$$\rho = 1 - \frac{6 \sum d_i^2}{n(n^2 - 1)}$$

where $d_i$ represents the difference between the ranks of corresponding variables $x_i$ and $y_i$, and $n$ is the number of observations. This metric offers insights into how well the ranking produced by the estimator preserves the monotonic relationship compared to the exact ranking, providing a measure of ranking fidelity.

Moreover, we not only consider the overall ranking but also evaluate the recovery of rankings for the top 20% of features. In scenarios with a large feature space, the most significant features often have a more pronounced impact on model predictions. In these cases, the overall ranking may be cluttered with a large number of features that show only minor differences in their Banzhaf values,

making it difficult to distinguish among lower-ranked features effectively. Focusing on the top 20% of features, therefore, targets those variables most likely to affect predictive accuracy and model stability, offering a more pragmatic evaluation of ranking recovery.

In our evaluation, we identify the top 20% of features ranked by exact Banzhaf values. For any feature missing from the top 20% as derived from the estimated Banzhaf values, we add it to the end of the permutation. This adjustment ensures that the evaluation penalizes discrepancies at the top of the distribution, which are most critical for decision-making and model interpretation.

Our results for overall feature ranking recovery are illustrated in Figure 11. When the feature size is small, our Kernel Banzhaf algorithm performs comparably to the MC method. However, as the feature size increases, the MC method shows superior performance. We hypothesize that this effectiveness stems from the MC method's precision in estimating zero Banzhaf values for non-contributory features, a key advantage when evaluating features individually in large datasets where many features have negligible Banzhaf values. It is important to note, however, that the MC method requires twice the number of samples and predictions compared to other methods, as it calculates the marginal contribution of each feature individually. This increased demand can make it less efficient, particularly with a large number of features.

Our findings for the top 20% feature ranking recovery are presented in Figure 12. This metric, focusing only on the ranks of the most significant features, is more practical. For datasets with feature size $\leq 20$, we take the top 7 features instead. Our Kernel Banzhaf algorithm consistently outperforms or matches the MC method across all datasets, demonstrating its effectiveness in identifying and ranking important features. The comparatively poorer performance of the MC method in the top 20% feature ranking supports our earlier hypothesis, as this metric involves a smaller subset of features and fewer evaluations of zero contribution instances.

By using these metrics, we provide a comprehensive analysis of how well Banzhaf estimators can not only approximate raw values but also preserve and recover the most critical aspects of feature importance in machine learning models.

## J    ADDITIONAL EXPERIMENTS ON ADVERSARIAL PERTURBATIONS

To further demonstrate the robustness of our proposed estimator, instead of independently perturbing all queries to the set function, we only perturb sets $S$ that contain a chosen item $i$. In the first experiment setting, we select $i$ uniformly at random. Then, instead of observing $v(S)$ on the query to subset $S$, the algorithms observe $v(S)+x$ where $x$ 0 if $i \notin S$ and, if $i \in S$, we have $x \sim \mathcal{N}(0, \sigma^2)$ as before. The results are as shown in Figure J.

Beyond this, in a follow-up experiment, we introduce a more adversarial noise setting. Each algorithm is run once on set function $v$ (no perturbation in the query access). Then we compute the relative error of each estimated value $\tilde{\phi}_j$ relative to the baseline $\phi_j$. We select the item $i$ with the largest relative error. Then, we evaluate each algorithm as before, but now the queries are perturbed if the set $S$ contains the adversarially chosen $i$. The results can be found in Figure 13

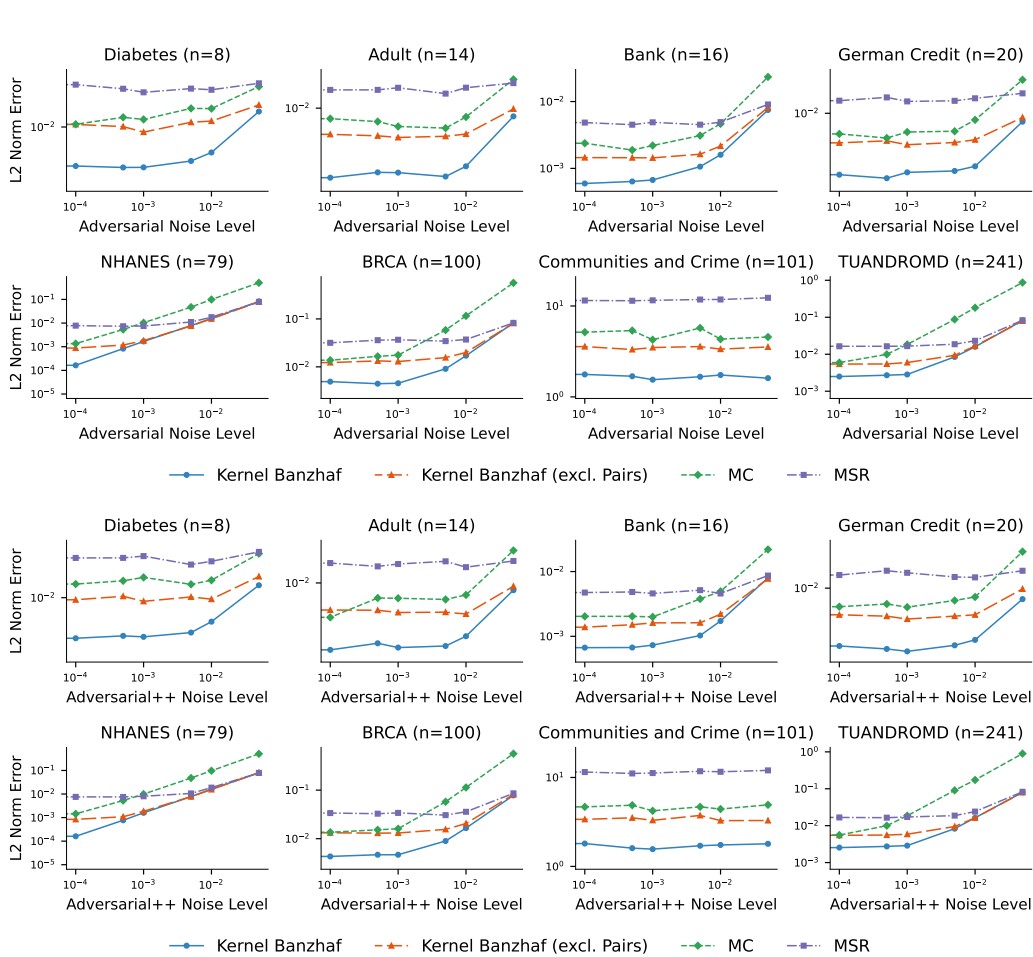

Figure 13: Plots of $\ell_2$-norm error by adversarial noise levels across Banzhaf estimators. For each noise level $\sigma$, the estimator observes $v(S) + x$ where $x \sim N(0, \sigma)$ when sets $S$ contains a randomly chosen feature $i$ (top panel), or feature $i$ with the largest relative error in the normal setting (bottom panel). Similar to previous experiments, Kernel Banzhaf outperforms for lower noise levels, eventually matching its ablated version and MSR for larger noise.

