# OpenReview forum: "Kernel Banzhaf: A Fast and Robust Estimator for Banzhaf Values"
_ICLR.cc/2025/Conference — Submitted to ICLR 2025_

### Official Review · Reviewer_DD24 · 2024-10-21

**Soundness:** 3
**Presentation:** 3
**Contribution:** 2
**Rating:** 6
**Confidence:** 4

**Summary:**

This paper proposes a new estimator for Banzhaf, which can be used to derive feature importance for general ML models. Theoretical analysis provides control over the error of the estimator.

**Strengths:**

* The proposed estimator seems to be more precise than current practice and hold theoretical guarantees.
* Experiments show that Kernel Banzhaf empirically has better sample complexity on eight tabular datasets.

**Weaknesses:**

* While the authors show that their approach achieves good sample complexity, it is unclear how meaningful that improvement is in practice from the current manuscript. I would make two suggestions: (1) can you use the proposed method to analyze datasets of large sizes in which MC and MSR fail to produce meaningful results but Kernel Banzhaf succeeds? (2) For the datasets you analyze, can you show that Kernel Banzhaf recovers feature ranking (overall and among the top-k features), or a similar quantity the practitioners would typically be interested in?
* This work is similar to Musco & Witter, and while there are differences (Banzhaf instead of Shapley, and the theoretical analysis required different techniques), the level of novelty in this work is not very high.

**Questions:**

* The MSR estimator should obtain sample complexity that is comparable to proposer method under the classification setting. How you explain the fact Kernel Benzhaf obtains better results in the experiments for the classification datasets? Is that true in general or not?
* Can't the theoretical results of Wang & Jia be extended to regression by normalizing the responses?
* In the contribution you write: "We argue that, up to log factors and the dependence on ϵ, our analysis is the best possible". What you mean by best? Do you mean tight? Or do you mean it is the best possible estimator for Banzhaf values?

---

> ### Author Response · Authors · 2024-11-20
>
> Dear Reviewer DD24,
>
> Thank you for your time and feedback! We respond to your concerns here, and your questions in another comment below.
>
> > can you use the proposed method to analyze datasets of large sizes in which MC and MSR fail to produce meaningful results but Kernel Banzhaf succeeds?
>
> Thank you for your suggestion! We have run an additional experiment on the large MNIST dataset; however, we find it challenging to define what constitutes "meaningful results”. For clarity, we focus on accurately estimating Banzhaf values in the quantitative sense of recovering the true Banzhaf values.
>
> The MNIST dataset consists of 784 features (28x28 pixels). In order to get quantitative results, we trained an XGBoost model on MNIST, which allows us to use TreeBanzhaf for calculating ground truth Banzhaf values. We then used the three estimators to estimate the Banzhaf values for 20 randomly selected images. We report the $\ell_2$-norm error at the 25%, 50%, and 75% percentiles when we use $m=10n$ samples as follows:
>
> |                    | 1st Quartile | 2nd Quartile | 3rd Quartile |
> |--------------------|--------------|--------------|--------------|
> | MC                 | 2.64         | 2.88         | 3.36         |
> | MSR                | 2.99         | 3.24         | 3.57         |
> | Kernel Banzhaf (excl. Pairs) | 2.61 | 2.86 | 3.27 |
> | Kernel Banzhaf     | **2.58**         | **2.81**         | **3.23**         |
>
> These results confirm the effectiveness of our proposed Kernel Banzhaf, both with and without paired sampling, when applied to image data with a large number of features.
>
> > For the datasets you analyze, can you show that Kernel Banzhaf recovers feature ranking (overall and among the top-k features), or a similar quantity the practitioners would typically be interested in?
>
> We appreciate the reviewer's suggestion to evaluate how our estimators recover feature rankings based on exact Banzhaf values, both overall and within the top-$k$ features setting. We have subsequently conducted these experiments, using Cayley distance and Spearman Correlation Coefficient as evaluation metrics. The results have been incorporated into **Appendix I** of our revised manuscript along with a detailed analysis. In the overall feature ranking experiment, Kernel Banzhaf outperforms MSR but MC gives the best performance. We suspect this is because MC can accurately recover Banzhaf values close to 0 (the average of $v(S \cup \{i\}) - v(S)$ are small for such Banzhaf values). However, in practice, we are less interested in the rankings of small Banzhaf values and would instead prioritize the rankings of large and important Banzhaf values. In the top-$k$ setting, we show that Kernel Banzhaf outperforms the other estimators, aligning with practical needs in prioritizing the most important features.
>
> > This work is similar to Musco & Witter, and while there are differences (Banzhaf instead of Shapley, and the theoretical analysis required different techniques), the level of novelty in this work is not very high.
>
> Leverage score sampling is a well known technique for sampling regression problems. The approach has been used since 2006 in work by Sarlos and others (e.g., see “Sketching as a Tool for Numerical Linear Algebra” by Woodruff for a good overview). To the extent that we use leverage score sampling to solve a regression problem, our work is similar to Musco & Witter (and others). However, the main contribution of our work remains novel:
>
> 1. The regression formulation of Shapley values has been known since the 80’s. In contrast, the analogous connection for Banzhaf values was only known for a special kind of set function up until our work. A significant portion of our contribution is framing Banzhaf values as a solution to a linear regression problem for arbitrary set functions.
>
> 2. Using this novel regression formulation, we design a sampling algorithm to estimate Banzhaf values. Because Kernel Banzhaf uses leverage score sampling (a common randomized linear algebra technique that is also used by Musco & Witter), our algorithm offers theoretical guarantees. The proof of these guarantees adapts the standard leverage score analysis to our sampling approach (and differs from the sampling without replacement in Kernel SHAP and Leverage SHAP).
>
> 3. Prior work on Banzhaf values estimation uses convergence stability as a measure of accuracy. We conduct extensive experiments across eight datasets where we compare estimated Banzhaf values to the *true* Banzhaf values. Kernel Banzhaf systematically outperforms prior work in these experiments.
>
> We hope this clarification underscores the novel contributions of our work.
>
> Due to space constraints, we respond to your questions in a comment below.

---

> ### Author Response · Authors · 2024-11-20
> **Answers to Questions**
>
> > How you explain the fact Kernel Benzhaf obtains better results in the experiments for the classification datasets? Is that true in general or not?
>
> While both MSR and Kernel Banzhaf have similar theoretical guarantees, the theoretical analysis does not exactly characterize the actual performance of the algorithms. In our experiments, we find that Kernel Banzhaf systematically outperforms MSR.
>
> We note that the theoretical guarantees of Kernel Banzhaf are actually stronger than those of MSR in the more general regression setting: The analysis of MSR in prior work assumes that the set function is bounded whereas our guarantees of Kernel Banzhaf are scale-invariant.
>
> > Can't the theoretical results of Wang & Jia be extended to regression by normalizing the responses?
>
> Wang & Jia analyze a sum of random variables under an assumption that each term is bounded. The Kernel Banzhaf estimator has a more complicated form i.e., $(\tilde{A}^T \tilde{A})^{-1} \tilde{A}^T \tilde{b}$. Showing this estimator is accurate requires showing the sampled matrix $\tilde{A}$ is close to the full matrix $A$ in both Frobenius norm and spectral norms, which we accomplish using matrix concentration inequalities and approximate block sampling analysis. We do not immediately see how to adapt their techniques to our setting.
>
> > In the contribution you write: "We argue that, up to log factors and the dependence on ϵ, our analysis is the best possible". What you mean by best? Do you mean tight? Or do you mean it is the best possible estimator for Banzhaf values?
>
> Good question! We mean that the approximation guarantee in Theorem 3.3—and Corollary 3.4 because they are equivalent—is nearly tight for regression-based algorithms like Kernel Banzhaf. It is possible that we can hope to do better because of the special structure of the Banzhaf regression problems; however, we suspect that this is not the case. A natural first step in showing the guarantee is nearly tight would be adapting the lower bound of Chen & Price 2019 for the structure of Banzhaf regression problems. We will make this clear in the final version of the paper.
>
> In terms of the best possible approximation for estimating Banzhaf values for any algorithm (not necessarily regression-based), $\Omega(n)$ is a natural lower bound. To see why, consider the following case: Suppose the set function can be written as $v(S) = \sum_{i \in S} w_i$ for some set of weights $w_1, \ldots, w_n$. In this setting, the Banzhaf values are exactly equal to $w_1, \ldots, w_n$. So, we must learn these weights exactly to learn the Banzhaf values. If we query $v(S)$ for fewer than $n$ subsets, we obtain a linear system with more unknowns than equations, so we cannot determine the values of $w_1, \ldots, w_n$. We suspect that the lower bound is actually closer to $\Omega(n/\epsilon)$, but we will have to think about how to show this!

---

> > ### Comment · Reviewer_DD24 · 2024-11-21
> > **thank you**
> >
> > Thank you for running these additional investigations, and it is good to see that there are some benefits in recovering feature rankings. I have increased my confidence to 4.

---

### Official Review · Reviewer_vb27 · 2024-11-01

**Soundness:** 4
**Presentation:** 4
**Contribution:** 3
**Rating:** 8
**Confidence:** 4

**Summary:**

In this paper, the authors proposed an efficient method for approximating the Banzhaf value. The Banzhaf value, similar to the Shapley value, is a measure used in cooperative game theory. Unlike the Shapley value, however, the Banzhaf value assigns equal weights to all subsets. The authors showed that the Banzhaf value can be represented as the solution to a least squares problem, and they propose a sampling-based approach to approximate this least squares solution. Through experiments, the authors demonstrated that their method achieves higher accuracy than other existing methods for approximating the Banzhaf value.

**Strengths:**

A key strength of this research is the simplicity of the proposed estimator for the Banzhaf value. The method involves simply sampling subsets and solving a least squares problem, making the computation highly straightforward. Additionally, the theoretical complexity of the sampling process is studied. While an exact calculation requires all the $2^n$ subsets, the proposed approach reduces this to approximately $O(n \log n / \delta)$. This ease of implementation, along with the theoretical guarantees, gives the study valuable for applications involving the Banzhaf value.

The discussion in Appendix H regarding the (un)necessity of efficiency axiom is particularly interesting. I think the efficiency axiom is not necessary within the context of feature attribution. Therefore, this discussion supporting the usefulness of the Banzhaf value is especially important.

**Weaknesses:**

There are no obvious weaknesses I found in this paper. If I have to mention a potential drawback, it might be that the Banzhaf value is less well-known compared to the Shapley value. However, as the authors discuss in Appendix H, the Banzhaf value can serve as a viable alternative to the Shapley value, and it would be ideal to see it become more widely studied alongside the Shapley value in the future.

**Questions:**

It is generally possible to achieve variance reduction by combining multiple estimators.
Would it be possible to create an estimator with lower variance by mixing the proposed method with MC and MSR estimators using appropriate weights?
If further variance reduction can be achieved, it would be highly useful for practical applications.

---

> ### Author Response · Authors · 2024-11-20
>
> Dear Reviewer vb27,
>
> Thank you for your time and feedback! We respond briefly below.
>
> > Banzhaf value is less well-known compared to the Shapley value. However, as the authors discuss in Appendix H, the Banzhaf value can serve as a viable alternative to the Shapley value, and it would be ideal to see it become more widely studied alongside the Shapley value in the future.
>
> We agree that Banzhaf values are a compelling alternative to Shapley values and we would also love to see more work in this area. We view Kernel Banzhaf as a valuable tool for the further study of Banzhaf values.
>
> > It is generally possible to achieve variance reduction by combining multiple estimators. Would it be possible to create an estimator with lower variance by mixing the proposed method with MC and MSR estimators using appropriate weights? If further variance reduction can be achieved, it would be highly useful for practical applications.
>
> Thank you for the insightful suggestion. Combining multiple estimators for variance reduction is indeed a promising approach. We have considered weighted mixing of estimators but did not explore it extensively in this paper. Your recommendation provides a valuable direction for future research.

---

### Official Review · Reviewer_W5vF · 2024-11-03

**Soundness:** 3
**Presentation:** 3
**Contribution:** 2
**Rating:** 5
**Confidence:** 3

**Summary:**

This work applies ideas proven effective for estimating Shapley values to Banzhaf values, introducing Kernel Banzhaf, a regression-based approximation algorithm for estimating Banzhaf values of general set functions. The authors demonstrate through extensive experiments that Kernel Banzhaf has significant advantages in sample efficiency and noise robustness. Additionally, they provide theoretical guarantees for the algorithm's performance.

**Strengths:**

1. Few algorithms have been proposed to compute Banzhaf values for arbitrary set functions. This paper addresses this gap by introducing an algorithm that overcomes this limitation, representing a significant improvement. It also experimentally evaluates the estimator in relation to the true Banzhaf values,rather than relying just on convergence metrics.

2. Theorem 3.2 states that the Banzhaf values are the solution to the linear regression problem defined by matrix A and vector b. Theorem 3.3 is a standard guarantee for leverage score sampling. Corollary 3.4 Kernel Banzhaf can recover a solution  that has near optimal objective value but is far from the optimal solution .

3. This work compared the Kernel Banzhaf with state-of-the-art estimators across eight popular datasets, and the results confirmed the superior performance of the Kernel Banzhaf.

**Weaknesses:**

1.While the theoretical underpinnings are well-developed, the paper may not provide a comprehensive assessment of the computational efficiency and practicality of the proposed method in real-world applications. Like the computational complexity analysis or empirical time/memory cost.

2.The study demonstrates the robustness of the Kernel Banzhaf algorithm primarily through relevant experiments. Figure 4 shows the horizontal line representing Kernel Banzhaf, which remains unchanged as noise levels increase.Previous studies, such as Data Banzhaf[1], have provided theoretical proof of robustness using the Safety Margin. This study may need to supplement related theoretical proofs.

**Questions:**

1.Broader baselines and empirical settings. For example, the settings for “Noisy” are kind of simple. What’s the variance of the added noise? The study claims to evaluate the Banzhaf values of general set functions and suggests expanding the dataset range to explore more scenarios, such as MNIST, FMNIST, and CIFAR-10.

Minor:
line106： What does  mean, and is it consistent with Data Banzhaf[1] ? Does it represent -approximation in -norm.

Ref.
[1]Jiachen T. Wang and Ruoxi Jia. Data banzhaf: A robust data valuation framework for machine learning. In AISTAT,  2023.

---

> ### Author Response · Authors · 2024-11-20
>
> Dear Reviewer W5vF,
>
> Thank you for your time and feedback! We respond to your concerns and questions below.
>
> > the paper may not provide a comprehensive assessment of the computational efficiency ... Like the computational complexity analysis or empirical time/memory cost.
>
> We analyze the time complexity of the proposed algorithm in lines 236-243 of the original paper. In particular, we show that Kernel Banzhaf runs in time $O(T_m +  mn^2)$, where $T_m$ is the time complexity to evaluate the set function $v$ on $m$ samples and $n$ is the number of features/observations. In most settings, we expect the time complexity of evaluating the set function to dominate (e.g., evaluating even a two-layer fully connected neural network requires $m$ passes with $O(n^2)$ time per pass). We confirm this experimentally in Figure 8 of Appendix E, which shows the empirical time cost as a function of the number of samples: For all estimators, the time complexity is dominated by evaluating the set function. We will make this analysis and experiment more clear in the final version of the paper.
>
> Please let us know if there are additional analyses or experiments that you would like us to run to shed additional light on the computational efficiency of Kernel Banzhaf.
>
> > …Previous studies, such as Data Banzhaf[1], have provided theoretical proof of robustness using the Safety Margin. This study may need to supplement related theoretical proofs.
>
> Thank you for this suggestion! In the Data Banzhaf paper, the primary task is to preserve rankings of observations hence the notion of safety margin naturally captures this goal. In our paper, the primary task is to accurately recover the true Banzhaf values. In this setting, a natural starting point may be analyzing the estimator error under Gaussian noise. We believe this is a promising direction for future work, but is outside the scope of our current work.
>
> > Broader baselines and empirical settings.
>
> We recognize the importance of comparing our approach with a broad range of baselines. Currently, our comparisons include Monte Carlo (MC) and Maximum Sample Reuse (MSR), which are the two methods used in prior work for approximating Banzhaf values. We also compare with state-of-the-art Shapley value estimators to demonstrate our method's efficiency and robustness. We welcome any additional suggestions for baselines that you believe could enhance our analysis.
>
> > For example, the settings for “Noisy” are kind of simple. What’s the variance of the added noise?
>
> In our robustness experiments (e.g., Figure 3), we add normally distributed noise to the set function. We explore different variances of this noise from the set [0, 0.0001, 0.0005, 0.001, 0.005, 0.01, 0.05]. We welcome any suggestions for additional robustness experiments.
>
> > The study claims to evaluate the Banzhaf values of general set functions and suggests expanding the dataset range to explore more scenarios, such as MNIST...
>
> Thank you for the suggestion of specific datasets from other application domains. In response, we have incorporated experiments using the MNIST dataset, which consists of 784 features (28x28 pixels). In order to get quantitative results, we trained an XGBoost model on MNIST, which allows us to use TreeBanzhaf for calculating ground truth Banzhaf values. We then used the three estimators to estimate the Banzhaf values for 20 randomly selected images. We report the $\ell_2$-norm error at the 25%, 50%, and 75% percentiles when we use $m=10n$ samples as follows:
>
> |                    | 1st Quartile | 2nd Quartile | 3rd Quartile |
> |--------------------|--------------|--------------|--------------|
> | MC                 | 2.64         | 2.88         | 3.36         |
> | MSR                | 2.99         | 3.24         | 3.57         |
> | Kernel Banzhaf (excl. Pairs) | 2.61 | 2.86 | 3.27 |
> | Kernel Banzhaf     | **2.58**         | **2.81**         | **3.23**         |
>
> These results confirm the effectiveness of our proposed Kernel Banzhaf, both with and without paired sampling, when applied to image data with a large number of features.
>
> > What does $\gamma$ mean, and is it consistent with Data Banzhaf? Does it represent $\ell_2$-approximation in $\ell_2$-norm.
>
> $\gamma$ is a parameter introduced in our theoretical analysis in the $\ell_2$-approximation factor of Kernel Banzhaf. Intuitively, $\gamma$ quantifies the quality of the optimal solution to the regression problem. We believe this parameter is fundamental to any regression-based approach for estimating Banzhaf values: Since Theorem 3.3 appears to be nearly tight (up to logarithmic factors in the sample complexity), and Corollary 3.4, which depends on $\gamma$, is equivalent to Theorem 3.3, it suggests that this dependence on $\gamma$ is also nearly tight. $\gamma$ does not appear in the analysis of the Data Banzhaf estimator MSR; however, we emphasize that $\gamma$ is small in practice and Kernel Banzhaf still systematically outperforms MSR.

---

> > ### Author Response · Authors · 2024-11-24
> >
> > Dear Reviewer W5vF02,
> >
> > Did our response address your concerns and questions? If not, we would love to carry out additional experiments and/or provide further clarification.

---

> > > ### Author Response · Authors · 2024-11-28
> > > **Additional Experiment with Adversarial Noise**
> > >
> > > Dear Reviewer W5vF02,
> > >
> > > > the settings for “Noisy” are kind of simple
> > >
> > > In response to your comment and that of another reviewer about the simplicity of the noise experiments, we have added two new experiments. In total, we now have three noise experiments with varying degrees of complexity and sophistication.
> > >
> > > **Experiment 1** (already in the paper): Instead of observing $v(S)$ on the query to subset $S$, the algorithms observe $v(S) + x$ where $x$ is drawn from a centered normal distribution with variance $\sigma^2$.
> > >
> > > The new experiments address the idea of more structured and adversarial noise.
> > >
> > > **Experiment 2** (new experiment): Instead of independently perturbing all queries to the set function, we only perturb sets $S$ that contain a chosen item $i$. In particular, we select $i$ uniformly at random. On the query to subset $S$ where $i \not \in S$, we observer $v(S)$. On the query to subset $S$ where $i \in S$, we observe $v(S) + x$ where where $x \sim \mathcal{N}(0, \sigma^2)$ as before.
> > >
> > > We run Experiment 2 for each dataset, each estimator, and each value $\sigma^2 \in$ {$0, 0.0001, 0.0005, 0.001, 0.005, 0.01, 0.05$} with 20 repetitions. **Appendix J contains plots of the results.** For ease of access, we present the median error when $\sigma^2=0.0001$ in the table below.
> > >
> > > | | Kernel Banzhaf | Kernel Banzhaf (excl. Pairs) | MC | MSR |
> > > |:-----------------------------|----------------:|-----------------------------:|-----:|------:|
> > > | Diabetes (n=8) | 0.0032 | 0.0108 | 0.0108 | 0.0347 |
> > > | Adult (n=14) | 0.0017 | 0.0051 | 0.0076 | 0.016 |
> > > | Bank (n=16) | 0.0006 | 0.0014 | 0.0024 | 0.0048 |
> > > | German Credit (n=20) | 0.0019 | 0.0045 | 0.0058 | 0.014 |
> > > | NHANES (n=79) | 0.0002 | 0.0009 | 0.0014 | 0.0078 |
> > > | BRCA (n=100) | 0.0049 | 0.0123 | 0.0137 | 0.0317 |
> > > | Communities and Crime (n=101) | 1.7677 | 3.5753 | 5.1517 | 11.4899 |
> > > | TUANDROMD (n=241) | 0.0025 | 0.0054 | 0.0059 | 0.0164 |
> > >
> > > **In this more structured and adversarial experiment (Experiment 2), Kernel Banzhaf continues to give the best performance.**
> > >
> > > We next test the estimators in an even more adversarial experiment described below.
> > >
> > > **Experiment 3** (new experiment): Each algorithm is run once on set function $v$ (no perturbation in the query access). We compute the relative error of each estimated value $\tilde{\phi}_j$ relative to the baseline $\phi_j$. We select the item $i$ with the largest relative error. Then, we evaluate each algorithm as before but now the queries are perturbed if the set $S$ contains the adversarially chosen $i$.
> > >
> > > We run Experiment 3 for each dataset, each estimator, and each value $\sigma^2 \in$ {$0, 0.0001, 0.0005, 0.001, 0.005, 0.01, 0.05$} with 20 repetitions. **Appendix J contains plots of the results.** For ease of access, we present the median error when $\sigma^2=0.0001$ in the table below.
> > >
> > > | | Kernel Banzhaf | Kernel Banzhaf (excl. Pairs) | MC | MSR |
> > > |:-----------------------------|----------------:|-----------------------------:|-----:|------:|
> > > | Diabetes (n=8) | 0.003 | 0.0094 | 0.0149 | 0.0323 |
> > > | Adult (n=14) | 0.0017 | 0.0049 | 0.004 | 0.0169 |
> > > | Bank (n=16) | 0.0007 | 0.0014 | 0.002 | 0.0047 |
> > > | German Credit (n=20) | 0.0021 | 0.0049 | 0.0061 | 0.0142 |
> > > | NHANES (n=79) | 0.0002 | 0.0008 | 0.0014 | 0.0076 |
> > > | BRCA (n=100) | 0.0043 | 0.0132 | 0.0136 | 0.0337 |
> > > | Communities and Crime (n=101) | 1.8072 | 3.3819 | 4.6692 | 11.5056 |
> > > | TUANDROMD (n=241) | 0.0025 | 0.0055 | 0.0056 | 0.0167 |
> > >
> > > **In this even more structured and adversarial experiment (Experiment 3), Kernel Banzhaf continues to give the best performance.**

---

> > > > ### Author Response · Authors · 2024-11-30
> > > >
> > > > Dear Reviewer W5vF,
> > > >
> > > > We realize that the end of the ICLR rebuttal phase is a particularly busy time! Nevertheless, we would appreciate your feedback on whether our response adequately addresses your initial concerns, or if there are any additional clarifications we can provide. Thanks again for your time!

---

### Official Review · Reviewer_BEq1 · 2024-11-24

**Soundness:** 3
**Presentation:** 3
**Contribution:** 2
**Rating:** 6
**Confidence:** 3

**Summary:**

Inspired by KernelSHAP, the paper introduces a method named "Kernel Banzhaf" that connects Banzhaf values to linear regression, leveraging "leverage score sampling" and "paired sampling" to approximate the Banzhaf values. The authors provide theoretical guarantees for the algorithm's performance and showcase its advantages in sample efficiency and robustness to noise through experiments on feature attribution tasks across eight datasets, outperforming existing estimators such as MC and MSR.

**Strengths:**

1. The paper is well-organized and clearly explains theoretical results and algorithms. In particular, I like that the authors kept the main paper simple while postponing the heavy theories and additional experiments and their analysis to the appendices.
2. Kernel Banzhaf addresses a gap in the computation of Banzhaf values for arbitrary set functions, an area with limited prior research compared to Shapley values.
3. The algorithm has solid theoretical support, as demonstrated by Theorem 3.2, Theorem 3.3, and Corollary 3.4, which ensure statistical accuracy and confidence and explain the connection to regression tasks. The authors also claimed that these results are "nearly optimal."

**Weaknesses:**

1. While the paper introduces a practical and efficient method for estimating Banzhaf values, much of its foundation relies on adapting existing techniques developed for Shapley values and generic regression problems.
2. Kernel Banzhaf demonstrates accuracy in Banzhaf value estimation, yet its broader implications for data valuation and generative AI tasks have not been explored. In particular, the authors consider that being inapplicable to generative AI is a limitation of MSR.
3. Robustness is primarily demonstrated through empirical evaluations, such as the $\ell_2$-norm error under varying noise levels (e.g., Figure 3). The paper does not explicitly incorporate noise-level assumptions and parameters into its theoretical guarantees (e.g., results in Section 3.3).

**Questions:**

1. As "Banzhaf values are often considered more intuitive for AI applications," is there a reason most existing studies focus on Shapley values?
2. How does Kernel Banzhaf perform under structured noise patterns, such as adversarial perturbations?

---

> ### Author Response · Authors · 2024-11-24
>
> Dear Reviewer BEq124,
>
> Thank you for your review!
>
> In terms of your question about unstructured noise, we would love to run an additional experiment in the "adversarial perturbation" setting you describe. What would such a setting look like? Please keep in mind that we only have two days to run this experiment because of when we received your review so we would very much appreciate a clarification soon.
>
> We will respond to your additional concerns and question below.
>
> > While the paper introduces a practical and efficient method for estimating Banzhaf values, much of its foundation relies on adapting existing techniques developed for Shapley values and generic regression problems.
>
> While we use regression sampling as in Kernel SHAP and Leverage SHAP, our work offers several novel contributions:
>
> 1. The regression formulation of Shapley values has been known since the 80's; however, for Banzhaf values, only a special case of this connection was known prior to our work. Formulating Banzhaf values as a solution to a linear regression problem is a key and non-trivial prerequisite to applying the regression-based algorithms used for Shapley values.
>
> 2. We apply leverage score sampling to the Banzhaf regression problem and exactly compute its leverage scores. Leverage score sampling is a well-studied technique that was recently applied to Shapley value estimation (Musco & Witter, 2024). In general, computing leverage scores is quite difficult. A large part of our contribution is exactly computing these values for the Banzhaf regression problem. We then apply variants of standard leverage score analysis to prove theoretical guarantees.
>
> 3. Prior work on Banzhaf value estimation used convergence as a proxy for accuracy. In our work, we exactly compute the Banzhaf values and compare the estimated values to these exact values. This results in a far more meaningful comparison, which we extend to 8 popular datasets, several natural hyperparameter settings, and the two Banzhaf value estimators used in prior work.
>
> > Kernel Banzhaf demonstrates accuracy in Banzhaf value estimation, yet its broader implications for data valuation and generative AI tasks have not been explored. In particular, the authors consider that being inapplicable to generative AI is a limitation of MSR.
>
> The focus of our work is on accurately estimating Banzhaf values in the general setting where the set function $v: \{0,1\}^n \to \mathbb{R}$ is unstructured. The point of our comment about MSR is to highlight that its theoretical guarantees require that $v: \{0,1\}^n \to [0,1]$ is bounded in a small interval. This means that the MSR guarantees are not applicable to regression tasks or potential generative AI applications. In contrast, because we make no assumptions on $v$, our guarantees can be used for any application of Banzhaf values. We believe this is particularly useful given that, as you say, Banzhaf values have been under-explored in the generative AI space. We leave the exploration of exactly how Banzhaf values can be used in generative AI to future work.
>
> > The paper does not explicitly incorporate noise-level assumptions and parameters into its theoretical guarantees (e.g., results in Section 3.3).
>
> This is an excellent suggestion! We believe this is a promising direction for future work, but is outside the scope of our current work. In particular, we feel that 1) formulating *general* Banzhaf values as a regression task, 2) designing a new algorithm for Banzhaf values using this connection, 3) exactly computing Shapley values for this regression task, 4) adapting standard analysis to prove theoretical guarantees for this algorithm, and finally 5) extensively evaluating Banzhaf approximation algorithms on the true Banzhaf values (contrasting with prior work) are already sufficient contributions.
>
> > As "Banzhaf values are often considered more intuitive for AI applications," is there a reason most existing studies focus on Shapley values?
>
> Shapley values are very popular in the literature. Part of their popularity is likely that they were adapted for explainable AI before Banzhaf values. One benefit of Shapley values is that they satisfy an "efficiency axiom" which means the Shapley values for each feature sum to the prediction of the model. One benefit of Banzhaf values is that they equally weight all subsets in their definition, leading to robustness and simplicity. Because of the benefits of Banzhaf values *and* their current underutilization, we believe our work is an important step towards understanding and computing these quantities.
>
> > How does Kernel Banzhaf perform under structured noise patterns, such as adversarial perturbations?
>
> Beyond the current noise experiments, we would be happy to run additional experiments. However, because your review was posted with only two days left in the discussion period, please quickly let us know what unstructured noise patterns experiments you'd like to see.

---

> ### Author Response · Authors · 2024-11-24
> **Additional Noise Experiments**
>
> In order to address your concerns about different types of structured noise, we are working on adding two new experiments. We describe the experiments below, and will post the results as soon as we have them (we're aiming for EOD tomorrow).
>
> Experiment 1 (already in the paper): Instead of observing $v(S)$ on the query to subset $S$, the algorithms observe $v(S) + x$ where $x$ is drawn from a centered normal distribution with variance $\sigma^2$.
>
> The new experiments address the idea of adversarial noise.
>
> Experiment 2 (new experiment): Instead of independently perturbing all queries to the set function, we only perturb sets $S$ that contain a chosen item $i$. In particular, we select $i$ uniformly at random. Then, instead of observing $v(S)$ on the query to subset $S$, the algorithms observe $v(S) + x$ where $x$ 0 if $i \not \in S$ and, if $i \in S$, we have $x \sim \mathcal{N}(0, \sigma^2)$ as before.
>
> The next experiment has noise that is even more adversarial.
>
> Experiment 3 (new experiment): Each algorithm is run once on set function $v$ (no perturbation in the query access). We compute the relative error of each estimated value $\tilde{\phi}_j$ relative to the baseline $\phi_j$. We select the item $i$ with the largest relative error. Then, we evaluate each algorithm as before but now the queries are perturbed if the set $S$ contains the adversarially chosen $i$.

---

> > ### Comment · Reviewer_BEq1 · 2024-11-25
> >
> > Thanks for the quick response. Experiment 2 should be good enough to show the algorithm's robustness under adversarial noise. If the time is insufficient, it would also be beneficial to simply include the above **descriptions** of these experiments in the appendices and briefly mention the newly added content in the main paper (e.g., as a proposal for future work).

---

> > > ### Author Response · Authors · 2024-11-28
> > > **Experiment Follow Up**
> > >
> > > Dear Reviewer BEq1,
> > >
> > > We just finished running the new noise experiments we discussed.
> > >
> > > In order to address your concerns about different types of noise, we have added two new experiments. The plots are in Appendix J and we show tables of the median error below. In total, we now have three noise experiments with varying degrees of complexity and sophistication.
> > >
> > > **Experiment 1** (already in the paper): Instead of observing $v(S)$ on the query to subset $S$, the algorithms observe $v(S) + x$ where $x$ is drawn from a centered normal distribution with variance $\sigma^2$.
> > >
> > > The new experiments address the idea of more structured and adversarial noise.
> > >
> > > **Experiment 2** (new experiment): Instead of independently perturbing all queries to the set function, we only perturb sets $S$ that contain a chosen item $i$. In particular, we select $i$ uniformly at random. On the query to subset $S$ where $i \not \in S$, we observer $v(S)$. On the query to subset $S$ where $i \in S$, we observe $v(S) + x$ where where $x \sim \mathcal{N}(0, \sigma^2)$ as before.
> > >
> > > We run Experiment 2 for each dataset, each estimator, and each value $\sigma^2 \in$ {$0, 0.0001, 0.0005, 0.001, 0.005, 0.01, 0.05$} with 20 repetitions. **Appendix J contains plots of the results.** For ease of access, we present the median error when $\sigma^2=0.0001$ in the table below.
> > >
> > > | | Kernel Banzhaf | Kernel Banzhaf (excl. Pairs) | MC | MSR |
> > > |:-----------------------------|----------------:|-----------------------------:|-----:|------:|
> > > | Diabetes (n=8) | 0.0032 | 0.0108 | 0.0108 | 0.0347 |
> > > | Adult (n=14) | 0.0017 | 0.0051 | 0.0076 | 0.016 |
> > > | Bank (n=16) | 0.0006 | 0.0014 | 0.0024 | 0.0048 |
> > > | German Credit (n=20) | 0.0019 | 0.0045 | 0.0058 | 0.014 |
> > > | NHANES (n=79) | 0.0002 | 0.0009 | 0.0014 | 0.0078 |
> > > | BRCA (n=100) | 0.0049 | 0.0123 | 0.0137 | 0.0317 |
> > > | Communities and Crime (n=101) | 1.7677 | 3.5753 | 5.1517 | 11.4899 |
> > > | TUANDROMD (n=241) | 0.0025 | 0.0054 | 0.0059 | 0.0164 |
> > >
> > > **In this more structured and adversarial experiment (Experiment 2), Kernel Banzhaf continues to give the best performance.**
> > >
> > > We next test the estimators in an even more adversarial experiment described below.
> > >
> > > **Experiment 3** (new experiment): Each algorithm is run once on set function $v$ (no perturbation in the query access). We compute the relative error of each estimated value $\tilde{\phi}_j$ relative to the baseline $\phi_j$. We select the item $i$ with the largest relative error. Then, we evaluate each algorithm as before but now the queries are perturbed if the set $S$ contains the adversarially chosen $i$.
> > >
> > > We run Experiment 3 for each dataset, each estimator, and each value $\sigma^2 \in$ {$0, 0.0001, 0.0005, 0.001, 0.005, 0.01, 0.05$} with 20 repetitions. **Appendix J contains plots of the results.** For ease of access, we present the median error when $\sigma^2=0.0001$ in the table below.
> > >
> > > | | Kernel Banzhaf | Kernel Banzhaf (excl. Pairs) | MC | MSR |
> > > |:-----------------------------|----------------:|-----------------------------:|-----:|------:|
> > > | Diabetes (n=8) | 0.003 | 0.0094 | 0.0149 | 0.0323 |
> > > | Adult (n=14) | 0.0017 | 0.0049 | 0.004 | 0.0169 |
> > > | Bank (n=16) | 0.0007 | 0.0014 | 0.002 | 0.0047 |
> > > | German Credit (n=20) | 0.0021 | 0.0049 | 0.0061 | 0.0142 |
> > > | NHANES (n=79) | 0.0002 | 0.0008 | 0.0014 | 0.0076 |
> > > | BRCA (n=100) | 0.0043 | 0.0132 | 0.0136 | 0.0337 |
> > > | Communities and Crime (n=101) | 1.8072 | 3.3819 | 4.6692 | 11.5056 |
> > > | TUANDROMD (n=241) | 0.0025 | 0.0055 | 0.0056 | 0.0167 |
> > >
> > > **In this even more structured and adversarial experiment (Experiment 3), Kernel Banzhaf continues to give the best performance.**

---

> > > > ### Comment · Reviewer_BEq1 · 2024-11-28
> > > >
> > > > Thanks for the effort! I'm impressed by this result - "In this even more structured and adversarial experiment (Experiment 3), Kernel Banzhaf continues to give the best performance."

---

> > > > > ### Author Response · Authors · 2024-11-28
> > > > >
> > > > > Thank you for your positive feedback on our additional results! If you have a chance, we’d love to hear your thoughts on whether our responses have addressed your concerns, or if there’s anything else we can add.

---

### Meta-Review · Area_Chair_zZKY · 2024-12-20

**Metareview:**

The paper proposes an algorithm called Kernel Banzhaf for estimating  Banzhaf values, which are an alternative to  Shapley values. The algorithm is inspired by KernelSHAP and leverages the connection between Banzhaf values and linear regression.Theoretical analysis and numerical experiments are provided.

Reviews are generally positive about the proposed algorithm and its analysis, however I share their concerns about its novelty and practical real world implications.

**Additional Comments On Reviewer Discussion:**

- Reviewer BEq1,  DD24: While the paper introduces a practical and efficient method for estimating Banzhaf values, much of its foundation relies on adapting existing techniques developed for Shapley values and generic regression problems.
In fact, while the authors claim to have adopted notation from (Musco and Witter, 2024), there appears to be substantial overlapping between the two papers in terms of both proof methodologies and results, with the Shapley values replaced by the Banzhaf values. In their rebuttal, the authors pointed out this difference. Since the  Shapley values and  Banzhaf values are quite similar, this reduces the novelty of the current work considerably.

- Reviewer BEq1: Kernel Banzhaf demonstrates accuracy in Banzhaf value estimation, yet its broader implications for data valuation and generative AI tasks have not been explored. In their rebuttal, the authors said they would explore this in future work. I suggest that this would make the contribution much stronger.

For these reasons, I find that the current contributions are not substantial enough to recommend acceptance.

Further note:

- The last equality in Eq(7) is not at all obvious. It should be considerably elaborated, as in the proof of Lemma 2.1 in (Musco and Witter, 2024).

---

### Decision · Program_Chairs · 2025-01-22

Reject